



# Deposition of brown carbon onto snow: changes of snow optical and radiative properties

Nicholas D. Beres[1,2], Deep Sengupta[1,2], Vera Samburova[1], Andrey Y. Khlystov[1], Hans Moosmüller[1]

[1]Division of Atmospheric Sciences, Desert Research Institute, Reno, NV, 89512, United States
[2]University of Nevada-Reno, Reno, NV, 89512, United States

*Correspondence to*: Nicholas D. Beres (Nic.Beres@DRI.edu)

**Abstract.** Light-absorbing organic carbon aerosol – colloquially known as brown carbon (BrC) – is emitted from combustion processes and has a brownish or yellowish visual appearance, caused by enhanced light absorption at shorter visible and
ultraviolet wavelengths ($0.3~\mu m \lesssim \lambda \lesssim 0.5~\mu m$). Recently, optical properties of atmospheric BrC aerosols have become the topic of intense research, but little is known about how BrC deposition onto snow surfaces affects the spectral snow albedo, which can alter the resulting radiative forcing and in-snow photochemistry. Wildland fires in close proximity to the cryosphere, such as peatland fires that emit large quantities of BrC, are becoming more common at high latitudes, potentially affecting nearby snow and ice surfaces.

In this study, we describe the artificial deposition of BrC aerosol with known optical, chemical, and physical properties onto the snow surface and we monitor its spectral radiative impact and compare it directly to modeled values. First, using small-scale combustion of Alaskan peat, BrC aerosols were artificially deposited onto the snow surface. UV-vis absorbance and total organic carbon (TOC) concentration of snow samples were measured for samples with and without artificial BrC deposition. These measurements were used to estimate the imaginary part of the refractive index of deposited
BrC aerosol with a volume mixing rule. Single particle optical properties were calculated using Mie theory, and these values were used to show that the measured spectral snow albedo of snow with deposited BrC was in general agreement with modeled spectral snow albedo using calculated BrC optical properties.

The instantaneous radiative forcing by impurities present in the snow before the deposition experiments was found to increase the instantaneous radiative forcing at the surface of the natural snow at our site by 1.23 (+0.14/-0.11) W m[-2] per ppm
of BrC deposited. However, we estimate that deposition onto a clean snowpack without light-absorbing impurities would have resulted in a more than twice as large instantaneous radiative forcing of 2.68 (+0.27/-0.22) W m[-2] per ppm of BrC deposited.

## 1 Introduction

Aerosol light absorption in the Earth's atmosphere lowers the planetary albedo, thereby causing radiative heating (Moosmüller et al., 2009; Satheesh and Krishna Moorthy, 2005; Stier et al., 2007; Stocker et al., 2013). Aerosols with a zero or very large
imaginary part of their refractive index do not absorb light (Moosmüller and Sorensen, 2018b, 2018a, 2019; Sorensen et al.,



2019). However, with the exception of metals, all atmospheric aerosols with a non-zero imaginary part of their refractive index do contribute to atmospheric light absorption. This absorption is distinct from other radiative forcing mechanisms such as aerosol scattering because light-absorbing aerosols continue to substantially contribute to radiative forcing after deposition onto high albedo surfaces such as snow and ice (Skiles and Painter, 2018; Warren, 1982). They contribute to climate forcing

(Hansen and Nazarenko, 2004; Jacobson, 2004) by significantly lowering snow and ice albedo (Chýlek et al., 1983; McConnell et al., 2007; Warren and Wiscombe, 1980), thereby reducing snow cover duration and changing spring runoff timing (Déry and Brown, 2007; Painter et al., 2007; Strack et al., 2007). Most previous work discusses reduction of snow albedo due to deposited black carbon (BC) and mineral dust, but little is known about snow albedo reduction due to deposited brown carbon (BrC). The light-absorbing properties of BrC – an optically defined component of organic carbon (OC) aerosol – in the

cryosphere are a growing topic of research. One source of atmospheric BrC is smoldering biomass combustion, which is becoming more common at high latitudes as peatlands increasingly dry out and become available as fuel for smoldering wildland fires (Brown et al., 2015; Kohlenberg et al., 2018; Turquety et al., 2007). Primary BrC aerosol deposition onto snow and ice is of particular concern in northern latitudes, due to the proximity of peat fuels to snow and ice surfaces (e.g., Evangeliou et al., 2019).

**1.1 Brown Carbon Aerosol**

Both BC and OC aerosols are formed by incomplete combustion of carbon containing fuels (Andreae and Gelencsér, 2006; Bond et al., 2013), and their contribution to a source's total emissions and their characteristics depends on many factors including fuel type, moisture content, packing density and source depth (Sumlin et al., 2018b), combustion phase (Bond et al., 2004; Patterson and McMahon, 1984; Reid et al., 2005), and other elements of the system (Chen et al., 2006). Many studies

have focused on the light absorption of the water-soluble fraction of OC, particularly that of humic-like substances (HULIS) (Graber and Rudich, 2006; Samburova et al., 2005; Sun et al., 2007). In addition, some recent work has focused on the light-absorption characteristics of non-water-soluble polycyclic aromatic hydrocarbons (PAHs) (Samburova et al., 2016) and other compounds derived from extractions with solvents of different polarity and pH (Sengupta et al., 2018).

OC compounds that are absorbing in the ultra-violet (UV) and short-visible wavelengths are colloquially known as

BrC. The wavelength dependency of BrC aerosol absorption coefficients can be described by an absorption Ångström exponent ($AAE$) that is significantly greater than one (Chakrabarty et al., 2010; Corr et al., 2012; Kirchstetter et al., 2004; McNaughton et al., 2011; Moosmüller et al., 2011), whereas BC aerosol absorption coefficients are less wavelength-dependent (i.e., $AAE \approx 1$) over the visible range and its periphery (Bergstrom et al., 2007; Moosmüller et al., 2009). The optical properties of BrC are described by the wavelength dependent BrC complex refractive index $m_\lambda$ written as

$$m_\lambda = n_\lambda + i\kappa_\lambda, \tag{1}$$





where $n_\lambda$ is the real part and $\kappa_\lambda$ is the imaginary part of the refractive index, the latter of which provides the primary control of the absorption coefficient (Sorensen et al., 2019). The imaginary part of the BrC refractive index (i.e., $\kappa_\lambda$) increases greatly toward shorter wavelength in the visible and near-UV spectrum, giving BrC its colored appearance and namesake (Andreae and Gelencsér, 2006). BrC aerosol optical properties are determined both by BrC bulk properties discussed above and by

particle size distribution and morphology. As BrC particles from smoldering combustion are mostly homogeneous (Sumlin et al., 2018a) and spherical (Chakrabarty et al., 2010), BrC aerosol optics can be described by BrC complex refractive index and size distribution using Mie theory (Mie, 1908; Moosmüller et al., 2011; Sumlin et al., 2018b).

A variety of fuels emit BrC during open combustion (Laskin et al., 2015), but peat is of particular interest for its common physical proximity to snow and ice at high latitudes of the northern hemisphere (Joosten and Clarke, 2002) and due

to its strong tendency to burn in the smoldering combustion phase (Watts and Kobziar, 2013). Peatlands are a land surface primarily found in the northern hemisphere composed of organic soil and decomposing plant material. They make up a small fraction (~2-3%) of Earth's land surface but store a significant amount (~25%) of the world's soil organic carbon (Turetsky et al., 2014; Xu et al., 2018). Under a warming climate, by the end of the 21$^{st}$ century, a doubling of boreal area burned per year has been predicted (Flannigan et al., 2009; Oris et al., 2014). Increases in boreal peatland fires will cause increased emissions

of OC and BrC aerosols at high latitudes.

Approximately 88% of all carbonaceous aerosol mass emitted globally is from wildland fires and biomass fuel combustion (Bond et al., 2004), and approximately 80% of those emissions are from smoldering combustion phase fires (Einfeld et al., 1991). OC is emitted together with BC from combustion sources, but more so in the smoldering phase (Bond et al., 2004; Patterson and McMahon, 1984; Reid et al., 2005) and it differs significantly in optical and chemical properties

(Bond et al., 2013; Chakrabarty et al., 2016; Kirchstetter et al., 2004; Laskin et al., 2015; Lewis et al., 2008; Moosmüller et al., 2009).

Carbonaceous particles also often undergo continued, secondary changes – physically, chemically, and optically – during transport within the atmosphere (Bhattarai et al., 2018; Jimenez et al., 2009; Sengupta et al., 2018; Sumlin et al., 2017). Chemical transformation of BrC during atmospheric transport can involve either fragmentation processes, producing lower

molecular weight compounds (Ervens et al., 2004) or oligomerization (Carlton et al., 2007).

**1.2 Brown carbon in the cryosphere**

OC from both natural and anthropogenic sources can be deposited onto snow after transport. The presence of OC in snow and ice not only plays a critical role in the carbon cycle of the cryosphere and of waterways downstream of snowmelt, but also contributes to the darkening of the snow surface at the short-visible and UV wavelengths. While studies of impurities in snow

and their reduction of snow albedo have focused on light-absorption by BC and/or mineral dust (Qian et al., 2015; Skiles et al., 2018), these impurities alone may not explain the decreased albedo at short-visible and UV wavelengths in the cryosphere.

Although only a small fraction of the total incoming solar irradiance at the snow surface is in the UV, it is a major driver of snow photochemistry that can be altered by BrC deposition near the top of the snowpack. Snow photochemistry





includes nitrate and nitrite photolysis, which releases $NO_x$ back into the atmosphere (Domine et al., 2008; Honrath et al., 1999, 2000) and mercury and bromine chemistry (Fain et al., 2006; Grannas et al., 2007). Additionally, biota in the cryosphere, such as snow algae (e.g., *Chlamydomonas nivalis*) (Thomas and Duval, 1995; Yallop et al., 2012), depend on incident light, and deposited impurities, such as BrC, can alter this interaction by absorbing light.

5      The influence of deposited BrC aerosols on surface and in-snow radiative properties is a recent and emerging topic. Grannas et al. (2007) provided an excellent overview of reactive organic compounds in snow and their impact on photolysis on and within the snow, but the direct impact of BrC on increased absorption and subsequent albedo effects was not addressed. Doherty et al., (2010) discovered the widespread presence of non-BC absorbing impurities (including ~25-50% contribution to total light-absorption by BrC) throughout the Arctic; the spectral albedo of these non-BC constituents was not reported. Similarly, Lin et al., (2014) extensively modeled the impact of BrC and other organic aerosols on global radiative forcing estimates. Wu et al., (2016) provided a nice but brief overview of BrC in the cryosphere and estimated effects on snow radiative forcing. One issue affecting advances in understanding the roles that BC and non-BC aerosols play for radiative forcing in and on snow is that partitioning their influence is difficult. Dang and Hegg (2014) described an extensive investigation identifying the absorbing components of non-BC materials – likely including BrC – in snow from across the Western United States using chemical extraction methods. For example, extraction of aerosols collected on filters allows for the separation of BC and non-BC constituents, but temporal resolution is lost when compared to real-time measurement methods.

Deposition of BrC onto and into snow and its radiative effects can be studied in detail and at sufficiently high BrC concentrations by artificially depositing BrC onto the snow surface (Beres and Moosmüller, 2018). Such experiments can deposit BrC from fuel sources, such as peat, that are likely to exist near snow surfaces, onto snow and quantify changes in snow composition (e.g., OC concentration) and radiative properties (e.g., spectral albedo). Here, we describe artificial deposition experiments, performed on natural snow surfaces in the Sierra Nevada, USA and report changes in snow OC content, meltwater absorption spectra, measured and modeled snow spectral albedos, and resulting radiative forcing due to the presence of impurities both naturally occurring and artificially deposited.

## 2 Methods

### 2.1 Study area and field work

An overview of methods used for this work is presented in Fig. 1 to guide through the relationships between measurements and modeling presented in the following sections. Experiments for this study were conducted at a small, alpine lake, Tamarack Lake, in the Carson Range of the Sierra Nevada in Nevada, USA. Depositions for this study took place during the winter/spring season of 2018-2019. Near the experiment site is a well-travelled mountain highway over Mount Rose pass, a ski resort, and the city of Reno, Nevada, USA. Sources of snow contamination include vehicle emissions along that highway, regional anthropogenic and biogenic sources in the Lake Tahoe region (Green et al., 2012; McDaniel and Zielinska, 2015), and long-range transport from California and Asia (Hadley et al., 2010). While our study site is not free from influences of natural and



anthropogenic sources, it provides a flat, easy-to-access, untrodden snow field, ideal for experiments involving the artificial deposition of aerosols. Experiments were conducted on a clear-sky, windless day.

Field experiments were conducted in the following order: (1) begin combustion of fuel to generate aerosols for deposition; (2) during deposition of aerosols (~40 min.), excavate abbreviated snow pit upwind of deposition, measure snow density, and collect snow samples of "natural" snowpack; (3) upon completion of aerosol deposition, immediately remove deposition apparatus and measure spectral albedo over the deposition area; (4) measure spectral albedo of "natural" snowpack several meters upwind of deposition area; (5) excavate snow pit at deposition site and collect snow samples. A second deposition experiment was conducted on the same day, and steps (1)-(5) were repeated and results included in this study.

Aerosol depositions onto the snow were made using the apparatus described by Beres and Moosmüller (2018). Briefly, combustion aerosols are produced and deposited onto and into the snowpack by means of two near-cylindrical volumes. The first acts as an aerosol production chamber, where small-scale combustion of fuels produces aerosols in-situ. Using an external, battery-powered pump, aerosols are pumped from the first to a second, larger volume that is placed over the snow surface where deposition is desired. Prior to combustion, the fuels are placed into a round, insulated container (volume of this container is ~1610 $cm^3$) to mimic real-world conditions in which surrounding, unburned peat provide insulation for the smoldering biomass, similar to the methods of Chakrabarty et al. (2016), Sengupta et al. (2018), and Sumlin et al. (2017, 2018b, 2018a). The fuels burned mostly with lower-temperature, smoldering-phase combustion producing OC-rich biomass-burning aerosols (any visible flaming combustion was quickly extinguished). Fuels burned for approximately 40 minutes, with air from the combustion volume being pumped into the deposition volume for an additional 10 minutes after the end of fuel combustion to encourage deposition of aerosols remaining in the volume. After this, the deposition volume was removed, and the experiment continued.

The fuel combusted for BrC aerosol deposition consisted of boreal peat samples collected from interior Alaska (hereafter "AK peat" or "AKP" in figures). Details of the fuel collection and preparation can be found in the supplemental information of Chakrabarty et al. (2016). The properties of this fuel's combustion emissions have been extensively studied under similar combustion conditions for their impact on air quality and radiative forcing in the atmosphere through optical, physical, and chemical characterization (Chakrabarty et al., 2016; Samburova et al., 2016; Sengupta et al., 2018; Sumlin et al., 2017, 2018a, 2018b). Peat fuels were harvested and stored under refrigeration after collection. A few days before snow deposition fieldwork, these fuels were conditioned in a heating and drying oven (Fisherbrand Isotemp, Waltham, MA) at a temperature of ~90 °C for one day in order to remove fuel moisture. Dried peat samples were weighed, and deionized water was sprayed homogeneously on those dry samples until the total mass of the wet peat samples showed a 25% increase compared to dried peat sample mass (i.e., fuel moisture content of 25%). Prior to their use in field combustion experiments, the wet peat samples remained for one day of equilibration in U-Line static shielding Ziploc bags.

Spectral albedo measurements of natural and aerosol-deposited snow surfaces were made using a fiber-coupled, high-resolution spectroradiometer (FieldSpec3, Analytical Spectral Devices, Inc., Boulder, CO, USA). This instrument measures over a wavelength range of 350-2500 nm with full-width, half maximum bandwidths of 3 nm (at 700 nm) to 10 nm (at 1400





and 2100 nm). Raw spectra are resampled and splined at 1 nm spectral resolution. Albedo was measured using a cosine-weighted optical diffuser at the fiber-end, mounted to a tripod for consistency of measurement location and angle. Reported albedo for this study is the ratio of the average of ten down-looking (upwelling radiation) measurements to the average of ten up-looking (downwelling radiation) measurements. We limit the reported albedo to the wavelength range of 350-1800 nm, due

to noise being introduced further in the NIR as well as the fact that aerosol absorption in snow mostly affects the UV and visible wavelengths albedo (Skiles et al., 2018; Warren and Wiscombe, 1980), which is the focus of this study. Spectral albedo of the deposited area was measured immediately after the deposition experiment, and spectra of an adjacent, undisturbed snowfield were collected immediately after that.

Snow samples were collected at each deposition area as well as at an upwind area to provide a background for OC

and other impurities present in the natural snowpack. First, samples were collected in an area of undisturbed natural snow, and later directly below the center of the deposition area at three depth ranges to gauge the depth at which BrC penetrated into the snowpack: surface (0 cm) to 2.5 cm (referred herein as "L1"), 2.5 to 7.5 cm ("L2"), and 7.5 to 12.5 cm below the surface ("L3") (Fig. 2). Each snow sample was placed into a polyethylene sampling bag (e.g., Whirlpak) and kept frozen in a freezer at -20 °C to prevent melting and minimize scavenging of OC.

**2.2 Laboratory measurements**

Light-absorbing impurities in snow include BC, mineral dust, and OC compounds; they are commonly quantified with optical methods (Cereceda-Balic et al., 2019; Grenfell et al., 2011; Skiles et al., 2017), X-ray fluorescence (Moosmüller et al., 2012), and chemical oxidation (Godec et al., 1990), respectively. Here, snow samples were processed to quantify total organic carbon (TOC) concentration and absorbance in the UV and visible wavelength ranges at the Desert Research Institute (DRI) in Reno,

NV, USA. For analysis, snow was transferred into 50-mL volume polypropylene vials that were first pre-soaked in 18.2 MΩ ultrapure water (UPW, model Purelab Ultra, Elga Labwater, Paris, France) for over 48-hours and then thrice rinsed using UPW. Snow was allowed to melt at room temperature (~21 °C) to discourage the volatilization of organics (as may occur when melting with a microwave or other accelerated means) and then immediately analyzed for TOC.

TOC was measured using a total carbon analyzer (Sievers 900-Series, SUEZ, San Jose, CA, USA). This instrument

measures total carbon (TC) and inorganic carbon (IC) concentrations and infers TOC as the difference of these two (i.e., TOC = TC – IC). TOC concentrations are reported as volume fractions (ppb) by the instrument but will be reported as OC mass concentrations herein (ppm = mg $L^{-1}$ = g $m^{-3}$) unless otherwise stated. According to the instrument manufacturer, a 60-μm stainless-steel inline filter is used to restrict very large plant detritus, insects, etc. from passing through the sample inlet line into the instrument. Because surface snow spectral albedo was measured with all particles present before and after the

deposition experiments, the snow meltwater samples were not filtered prior to TOC determination and therefore may contain particulate and non-particulate OC, as well as water-soluble and insoluble OC compounds. BrC aerosols generated through similar small-scale smoldering combustion experiments generally do not exceed 1 μm in diameter and have a geometric mean diameter close to 100 nm (Chakrabarty et al., 2016; Sumlin et al., 2017, 2018b) and therefore are included in the measured



TOC concentration. Contribution of the Whirlpak bags to measured TOC concentration is nonzero, even for samples that remain frozen (Legrand et al., 2013); however, we expect these discrepancies to mostly subtract out between deposited and natural snow samples. Contamination of the polypropylene vials with TOC was measured to be $0.020 \pm 0.001$ g m$^{-3}$, which was subtracted from each of the meltwater TOC measurements. For this study, we assume that while the natural snowpack

contributes to the measured TOC, it is free from influence of the aerosol deposition experiment. We can thus infer the background TOC concentration of both the natural snowpack located at the Tamarack Lake site as well as the direct TOC contribution of the BrC aerosol deposition experiment. That is, the OC added to the snowpack through the deposition experiment $TOC_{BrC,i}$ can be calculated for layer $i$ as

$$TOC_{BrC,i} = TOC_{bulk,i} - TOC_{snow,i},\qquad(2)$$

where $TOC_{bulk}$ is the measured TOC from the deposition site samples that include both OC already in the "natural" snowpack and artificially deposited OC and $TOC_{snow}$ is the measured TOC from the natural snowpack. The uncertainties presented in this study for TOC concentrations represent the standard deviation of the mean for all $n$ individual TOC determinations ($13 \leq$

$n \leq 72$) and $N$ replicate meltwater TOC measurements ($2 \leq N \leq 4$).

Light absorbance of the melted snow was measured using a Perkin-Elmer Lambda 1050 UV/Vis spectrophotometer (Waltham, MA, USA). Prior to measurement, the melted snow samples were sonicated for ~20 min. and stirred to ensure the solubility of water-soluble organic compounds (WSOC). However, uniformity of suspended particles cannot be controlled through sonication. Sonicated melt water samples were placed into a 1-cm-path-length cuvette. To avoid potential

contaminations between samples, this cuvette was flushed with methanol and rinsed several times with UPW. Measurement scans were performed for each sample over the 200-860 nm wavelength range with 1-nm resolution and with UPW serving as reference. The instrument reports the absorbance $A$ as the power exponent of the Beer-Lambert law, where the transmittance, $T$, of light through a solution is a ratio of the radiant flux transmitted $\Phi_e^t$ to the total radiant flux incident $\Phi_e^i$ on the sample. This method assumes there is little contribution of scattering to the overall extinction of light along the path (Bosch et al.,

2014). Transmittance $T$ is related to absorbance $A$ as

$$T = \frac{\Phi_e^t}{\Phi_e^i} = 10^{-A} = e^{-\tau},\qquad(3)$$

and absorbance $A$ is related to optical depth $\tau$ by

$$A = \frac{\tau}{\ln 10}.\qquad(4)$$





If the optical attenuation is uniform along the path of light, the absorption coefficient $\beta_{abs}$ is related to the optical depth $\tau$ in Eq. (4) as

$$\tau = l \cdot \beta_{abs}, \tag{5}$$

where $l$ is the path length of light through the liquid sample. For absorbance spectra, we consider the bulk matter of melted pure ice grains and the impurities that lie both internal and external to the grain matrix and evaluate them as a bulk sample. The bulk absorption coefficient $\beta_{abs\_bulk}$ is directly related to the imaginary refractive index $\kappa_{bulk}$ (Moosmüller et al., 2009) as

$$\beta_{abs\_bulk} = \frac{4 \cdot \pi \cdot \kappa_{bulk}}{\lambda}, \tag{6}$$

so, the estimation of $\kappa_{bulk}$ follows as

$$\kappa_{bulk} = \frac{\ln(10) \cdot \lambda \cdot A}{4 \cdot \pi \cdot l}. \tag{7}$$

The raw absorbance measurements were baseline-corrected to account for drift by subtracting the scan average over the $700 - 860$ nm wavelength range from individual wavelength absorbance values, similar to the method outlined by Sengupta et al., (2018). This also reduces the influence of any BC particle absorption that is fairly independent of wavelength across the

UV-vis spectrum, while only minimally affecting BrC particle absorption that is much greater at blue and near-UV wavelengths (Bahadur et al., 2012; Chakrabarty et al., 2010; Kirchstetter and Thatcher, 2012; Lu et al., 2015; Sumlin et al., 2018b). Additionally, each spectrum was smoothed using a running average of 15 spectral data points to reduce high-frequency noise while maintaining low-frequency structure.

Similar to measurements of TOC, we consider the bulk snow-aerosol sample when estimating a value for $\kappa$ – both

for the natural snowpack including its naturally-occurring impurities ($\kappa_{snow}$) and for the BrC aerosol deposited snow ($\kappa_{bulk}$) – as determined through spectrophotometric absorbance measurements. We can isolate the influence of BrC absorption (via the imaginary refractive index) by incorporating the TOC concentration through a simple volume mixing approximation (Chýlek et al., 1988) as

$$\kappa_{bulk,i} = \kappa_{BrC,i} \cdot VF_{BrC,i} + \kappa_{snow,i} \cdot VF_{snow,i}, \tag{8}$$

where $VF_{BrC}$ and $VF_{snow}$ are the volume fractions of BrC and bulk meltwater TOC, respectively, for each layer $i$, such that $VF_{BrC} + VF_{snow} = 1$. Because the absorbance measurements are made against UPW, we do not include $\kappa$ or the volume



fraction of UPW in Eq. (8). Therefore, the average imaginary part of the refractive index of BrC, here labelled as $\kappa_{BrC,i}$ for each layer $i$, can be written as

$$\kappa_{BrC,i} = \frac{\kappa_{bulk,i} - (\kappa_{snow,i} \cdot VF_{snow,i})}{VF_{BrC,i}}. \tag{9}$$

Because two samples of the natural snowpack at Tamarack Lake were collected on the same day, we average values of TOC and absorbance over these two natural snowpack samples for each layer. Henceforth, the natural snowpack will be presented as one set of three depths/layers unless otherwise stated.

## 3 Results and Discussion

Aerosol deposition experiments were conducted during April 2019. Two separate depositions of BrC aerosols produced from the smoldering combustion of AK peat were performed (Experiment 1 and 2), and the analysis of the snow properties and spectral albedo from the field and laboratory and modeling results are discussed below.

### 3.1 Field work

Two aerosol depositions were performed on April 24, 2019 (approximately two weeks after the last snowfall) at Tamarack
Lake. Snowpack and experimental conditions are presented in Table 1. The average snowpack depth measured on April 24, 2019 at Tamarack Lake was 180 cm. Liquid water was noted in approximately the top one cm of snow, but snow below the surface did not contain liquid water.

The first and second deposition included aerosol generation through smoldering combustion for 40 and 37 minutes, respectively, followed by 10 minutes of residual aerosol deposition. At the start of the experiments, the wet fuel mass was 55
g and 45 g for first and second deposition, respectively. The fuel packing density for both burns was ~0.03 g cm$^{-3}$, which is considered to be a low fuel-packing density (Sumlin et al., 2018b).

Similar to the BrC aerosol deposition experiment discussed by Beres and Moosmüller (2018), deposited BrC aerosol from the smoldering combustion of AK peat appeared yellowish to the eye, indicating a preferential light absorption of blue-violet wavelengths by the aerosol deposited. This strong wavelength dependence was quantified by the measured spectral
albedo for each deposition experiment, which is shown in Fig. 3 alongside measured albedo of the natural snowpack upwind of the deposition. Spectral albedo reduction due to the presence of freshly deposited BrC aerosol increased with decreasing wavelength in the UV-visible wavelength region for both deposition experiments. In the UV, there was a stronger reduction than in the visible, up to ~0.14 and ~0.21 at 350 nm for Experiment 1 and 2, respectively (Fig. 3). At 700 nm however, albedo for each deposition was reduced by less than 0.01, indicating very little BC added to the snowpack through the deposition
experiments. Both depositions reduced the spectral albedo in the measured near-infrared (NIR) region as well and we



hypothesize that the reduction of spectral albedo in the NIR is not due to the BrC or BC deposited but may be due to the deposition volume enclosing the snowpack and increasing temperature and accelerating metamorphism of the snow grains to larger grain sizes (Davis et al., 1993; Doherty et al., 2013), and thus increasing absorption in that spectral region (Dozier and Painter, 2004; Warren, 1982; Wiscombe and Warren, 1980). At 1030 nm, the albedo was reduced by 0.029 and 0.022 for the first and second experiments, respectively. Using SNICAR (see Sect. 3.2.2 and Flanner et al., 2007) to estimate effective grain radii by fitting modeled and measured albedos at that wavelength resulted in an increase of grain size from ~682 to ~807 μm and ~605 to ~695 μm for the deposition experiments 1 and 2, respectively.

The Sierra Nevada snowpack generally receives sufficient aerosol BC and non-BC absorbing aerosols to affect the energy balance in an appreciable manner (Dang and Hegg, 2014; Hadley et al., 2010; Sterle et al., 2013). The albedo of the natural snowpack measured during this study indicates the presence of absorbing impurities without artificial deposition of BrC: a hypothetical pure snow spectral albedo – holding other input parameters constant, modeled through SNICAR – would have an albedo near 0.98 at 350 nm. However, measured albedos at 350 nm for the natural snowpack are appreciably lower, near ~0.84. Along with BrC already present in the snow, other impurities are likely a mix of mineral dust and BC (Hadley et al., 2010; Sterle et al., 2013); however, it is BrC and mineral dust that are responsible for greater albedo decreases (compared to pure snow) at these lower wavelengths (Laskin et al., 2015; Lu et al., 2015; Moosmüller et al., 2009; Skiles et al., 2017; Warren et al., 2019; Wu et al., 2016). While we can account for the presence of BrC already in the snowpack by assigning the laboratory-measured UV-vis absorption to the measured TOC concentrations of the natural snowpack samples, dust concentrations were not measured and therefore remain unknown. However, they are subtracted out in our quantification of the imaginary part of the deposited BrC refractive index ($\kappa_{BrC}$) because we consider only the difference between natural snow with and without artificial deposition of BrC.

## 3.2 Analysis of snow samples

TOC concentrations in the snowpack before and after deposition are presented in Table 2. Averaging samples collected at both natural snow sampling sites, L1 TOC concentration was $0.579 \pm 0.014$ g m$^{-3}$, for L2 it was $0.436 \pm 0.048$ g m$^{-3}$, and for L3 it was $0.425 \pm 0.008$ g m$^{-3}$. Our OC concentrations fall within the wide range of values found in literature. For example, our concentrations are similar to those reported by Zhang et al. (2019) (and references therein) for OC in snow. Meinander et al. (2013) found a factor of six larger, where concentrations are heavily influenced through deposition of air pollution from mining and refining industry. Legrand et al. (2013) showed that values of OC found in pristine areas of the world are one or two orders of magnitude lower than our values. The deposition of BrC aerosol added at least one order of magnitude more TOC than was already in the snowpack and increased TOC concentrations below the surface layer as well. Experiment 2 deposited approximately twice the TOC mass concentration than Experiment 1, which is evidenced not only in the measured TOC concentration for each layer, but also in a stronger decrease of measured UV albedo for Experiment 2 than for Experiment 1, relative to the natural snow (Fig. 3). Therefore, deposited TOC greatly dominated that already existing in the natural snowpack. While the surface layer (L1) captured a majority of the deposited TOC, the BrC penetrated deeper into the snowpack during



the deposition experiments, either through movement or "pumping" of air into the snow (Colbeck, 1997; Harder et al., 1996; Waddington et al., 1996) or through melt-induced movement of soluble material through the snowpack (Meyer and Wania, 2008, and references therein). The latter may be more likely in a natural scenario of dry deposition (Clifton et al., 2008), although under the conditions of artificial deposition in the winter-spring transition period, the result may be due to both

pumping and meltwater flush.

After determining $\kappa_{bulk}$, we calculated the absorption coefficient $\beta_{abs\_bulk}$ following Eq. (6). The values of $\beta_{abs}$ for deposited BrC, $\beta_{abs\_BrC}$, are found by subtracting the natural snowpack meltwater absorption coefficient averaged over the two natural snowpack sampling sites, denoted as $\beta_{abs\_snow}$, from that of the deposited BrC-snow meltwater, denoted as $\beta_{abs\_bulk}$; spectra calculated for $\beta_{abs\_BrC}$ and $\beta_{abs\_snow}$ can be found in Fig. 4. Indicative of BrC in water and aerosol,

absorption spectra of meltwater from all samples exhibited a general increase in absorption with decreasing wavelength through the visible wavelength range, as well as peaks in the UV. $\beta_{abs\_snow}$ spectra had a local absorption peak near ~258 nm. UV absorption band in presence of an aromatic nuclei (no other functional group specified) can be around 255 nm, which is very close to our observed $\beta_{abs\_snow}$ values. The presence of aromatic nuclei in the absorption spectra may be attributed to the traffic emissions near the field site or to decomposition of indigenous biogenic compounds in the atmosphere and snow surface.

With the deposition of BrC aerosol to the snowpack, the absorption increased significantly in the short-vis and UV wavelength ranges and created a more prominent local absorption maximum at 275 nm, with $\beta_{abs\_BrC}$ spectra demonstrating a very similar pattern to phenolic compounds, ubiquitous in biomass combustion (Yee et al., 2013), with two peaks around 210 nm and 275 nm and a shoulder around 240nm (Linstrom and Mallard, 2018). The preponderance of phenolic compounds in absorption spectra over other chemical classes make them potential markers for BrC deposition on snow. However, the fate of such

phenolic compounds on snow surface, through photochemical processing, for example, is still unknown and should be further investigated.

The calculated absorption coefficients correlate positively with measured TOC mass concentration of these samples. The slope of the linear regression between absorption coefficient (m$^{-1}$) at wavelength $\lambda$ and TOC concentrations give the (mass) specific absorbance, $B_\lambda$ (m$^2$ g$^{-1}$), which is an indicator of the contribution by OC to the absorption coefficient in snow and ice

(Warren et al., 2019; Zhang et al., 2019). $B_\lambda$ – sometimes referred to as specific absorbance – at various wavelengths is used to describe other properties, such as aromaticity of the OC (Hansen et al., 2016), drinking water quality (Potter and Wimsatt, 2012), or used to help describe different processes in rivers, lakes, and oceans (Fichot and Benner, 2011; Twardowski et al., 2004; Yacobi et al., 2003). The $B_\lambda$ spectrum over the wavelength range of 215-815 nm is presented in Fig. 5 along with the calculated correlation coefficient $R^2$ for each regression using $TOC_{BrC}$ concentrations together with those of the natural

snowpack, $TOC_{snow}$, as one dataset. The absorption is well-explained ($R^2 > 0.9$) by TOC in the meltwater throughout the UV, but above 353 nm, the confidence in that relationship drops quickly ($R^2 < 0.5$). If we consider just the natural snow samples, the calculated $R^2$ doesn't suggest high confidence ($R^2 < 0.7$ across all wavelengths), likely due the low sample size (n=6). For the natural snow samples, $B_\lambda$ is nearly equivalent to that of BrC at a wavelength of 258 nm ($B_\lambda \approx 7.3$). $B_\lambda$ for the natural



snowpack is two orders of magnitude greater than values inferred by Warren et al. (2019) for Alaskan sea ice at 400 nm, but closer to that of snow and ice in the northern Tibetan Plateau (Yan et al., 2016).

The absorbance of BrC generally decreases with increasing wavelength, and a power-law relationship is often used to describe this wavelength dependence, as

$$p(\lambda) = c\lambda^{-AE},\qquad(10)$$

where $p$ is the parameter exhibiting the wavelength dependence, $c$ is a constant, $AE$ is the Ångström exponent (Ångström, 1929). For absorption, $p$ becomes the absorption coefficient $\beta_{abs}$, yielding the absorption Ångström exponent ($AAE$). The $AAE$ can be written for two wavelengths as (Moosmüller et al., 2011)

$$\frac{\beta_{abs}(\lambda_1)}{\beta_{abs}(\lambda_2)} = \left(\frac{\lambda_1}{\lambda_2}\right)^{-AAE},\qquad(11)$$

$$AAE_{\lambda_2}^{\lambda_1} = -\frac{\ln(\beta_{abs}(\lambda_2)) - \ln(\beta_{abs}(\lambda_1))}{\ln(\lambda_2) - \ln(\lambda_1)}.\qquad(12)$$

Zhang et al. (2019) have compiled a graphic (their Fig. 6) showing $AAE_{400}^{330}$ values for several studies of OC in snow and the atmosphere. Our values for $AAE_{400}^{330}$ (2.68, 1.84, and 1.48 for L1, L2, and L3, respectively) derived from the average natural snowpack $\beta_{abs\_snow}$ are on the lower-end of their spectrum for $AAE$s of snow, close to Zhang et al. (2019) results of OC in snow from the Altai Mountains and to that of Arctic snow (Doherty et al., 2010). $AAE_{400}^{330}$ derived from $\beta_{abs\_bulk}$ in our study has a range from 4.12 to 6.28 (mean = 5.32) for all layers; and, when considering $\beta_{abs\_BrC}$, the larger $AAE_{400}^{330}$ values are in the range of 4.86 to 9.79 (mean = 7.76). Our values for BrC-only $AAE$ over this wavelength range agree well with those presented in Fig. 6 of Zhang et al. (2019) for BrC aerosol in different regions of the world.

### 3.2.1 Estimation of the imaginary refractive index, $\kappa$

Equation (7) was used to estimate the imaginary part of the refractive index, $\kappa$, from each absorbance spectra. The volume mixing rule described in Eqs. (8, 9) was used to normalize the calculated $\kappa$ for each spectrum using the BrC volume fraction measured for each sample to obtain $\kappa_{BrC}$, the imaginary part of the BrC refractive index. The value of $\kappa_{BrC}$ displayed in Fig. 6 represents that for all samples containing deposited BrC from AK peat combustion. The black dotted lines represent the range of retrievals for $\kappa_{BrC}$ for each snow sample from the deposition experiments, which incorporate the uncertainty of TOC concentrations assigned to each absorption spectrum of BrC.

Comparing the values of $\kappa_{BrC}$ obtained with this method to those of other selected studies for peat biomass combustion shows general agreement, with differences being greatest at different wavelength regions for different studies.





Sumlin et al. (2018b) identified values of $\kappa_{BrC}$ at four wavelengths (375, 405, 532, and 1047 nm) under varying conditions of combustion for different peat fuels, including Alaskan peat; they identified the fuel packing density as the parameter dominating $\kappa_{BrC}$, but found little dependence on the type of peat. Sengupta et al. (2018) estimated the refractive index of BrC from smoldering combustion of peat from Siberia (SBP, unpublished results) and Florida (FP), extracted in water. $\kappa_{BrC}$ for
SBP decreased with increasing wavelength faster than that from other peat samples in Fig. 6. In the visible part of the spectrum, the values of $\kappa_{BrC}$ for FP smoke are in agreement with our results for AK peat smoke, while in the UV, the largest qualitative difference is the presence of a peak at 275 nm in our spectrum; this peak is not discernible in the other spectra.

### 3.2.2 Estimating changes in snow albedo and radiative forcing by BrC deposition

Using a complex index of refraction and particle size distribution as an input to Mie theory, we estimate single-particle optical
properties of BrC under the assumption of homogeneous, spherical particles. These properties are inputs characterizing impurities for the SNow, ICe, Aerosol, and Radiation (SNICAR) model (Flanner et al., 2007) to estimate spectral snow albedos over the wavelength range of 0.3 μm to 5.0 μm, enabling us to compare them to measured albedos of the artificially-deposited BrC discussed in Sect. 3.1. We limit the reported albedo comparison to the wavelength range of 350 – 845 nm to match measured spectral albedo using BrC optical properties derived through UV-vis spectroscopy.
15       Sumlin et al. (2018b) show that for BrC from combustion of AK peat, the real part of the refractive index, $n$, is insensitive to changes in fuel moisture content, source depth, and geographic origin, and is constrained between 1.5 and 1.7. Because of this and the fact that no information exists regarding $n$ below 375 nm or above 1047 nm for AK peat, we use $n$=1.6 across the wavelength range of interest and, along with the range of values retrieved for $\kappa_{BrC}$ in Sect. 3.3, as input for Mie theory calculations. We also utilize the Sumlin et al. (2018b) particle size distribution that was measured under similar
combustion conditions as encountered in our study for AK peat with low fuel packing density and smoldering combustion for 35-40 minutes. This lognormal distribution is described by a geometric mean diameter and standard deviation of 157 nm and 1.7, respectively. Mie theory calculations with our range of $\kappa_{BrC}$ used as input into the Mie code return a range of single particle properties for the single scattering albedo (SSA), asymmetry parameter (g), and mass extinction coefficient (MEC, m$^2$ kg$^{-1}$). The average BrC SSA returned increases with increasing $\lambda$ close to a value of 1.0 near $\lambda$ = 800 nm. The SSA values in
the UV and short visible wavelengths agree with those given by Sumlin et al. (2018b) in the UV and visible wavelength ranges, although they display the largest relative range of returned values from the Mie calculations, spanning 0.72 – 0.86 at 305 nm.

The widely-used SNICAR model (Flanner et al., 2007) uses a two-stream radiative transfer scheme to simulate the optical properties of snow and ice in the presence of different types of impurities under different solar and snowpack conditions. The model user can specify multilayer snowpack properties, including individual layers, their snow densities and effective
grain radii, as well as the concentrations of impurities, such as: BC, an absorbing mineral dust in four size ranges, and/or volcanic ash. Impurity concentrations are specified for each individual layer, and impurities are treated as externally mixed with the ice grains. Optical properties (SSA, g, and MEC) for all impurities are described in lookup tables. Here, we incorporate





our calculated SSA, g, and MEC for AK peat BrC into SNICAR and assign a BrC concentration in the modeled multi-layer snowpack equal to that measured by TOC analysis.

For this study, the modeled albedo was first matched to measured spectral albedo of the natural snowpack along the UV-visible wavelength range, considering both the observed snowpack properties and solar geometry for the measurement day and time. The average TOC concentration for the natural snow layers (L1, L2, and L3) is input as BrC concentration, and values for BC and mineral dust are added to match the modeled albedo to the measured albedo. Once reasonable agreement ($\Delta$albedo $\leq$ 0.025) is achieved for the natural snowpack measured and average modeled albedos across the wavelength range, the solar geometry, snowpack properties, and BrC concentration are updated to reflect that of the BrC deposition; results are presented in Fig. 7. Using this method, the difference in spectral albedo along the wavelength range in question was less than 0.15 for the deposited BrC for both deposition experiments. The modeled albedo for Experiment 1 generally matched the measured albedo better than Experiment 2, especially when considering the upper and lower bounds of BrC optical properties (shaded regions in figures). However, for Experiment 2, the additional deposited BrC resulted in an averaged modeled spectral albedo significantly lower (up to $\Delta$albedo = 0.063) than the measured albedo throughout most of the visible and short-visible wavelength ranges. For Experiment 1, the modeled albedo in the NIR for the natural snowpack did not match that of the measured albedo ($\Delta$albedo < 0.025) without adding more BC and removing the mineral dust concentration nearly completely, which is unrealistic for the Sierra Nevada snow during the winter-spring transition. Additionally, adding more BC would have lowered the albedo across the entire wavelength range of interest, not just in the NIR.

Because spectral albedo was measured before and after BrC aerosol deposition, we can directly assess the enhanced absorption of solar radiation from the deposited BrC by estimating the increase in instantaneous radiative forcing (RF) due to the presence of light-absorbing impurities as

$$RF = \sum_{\lambda_1}^{\lambda_2} I \Delta a \, \Delta\lambda, \tag{13}$$

where $I$ is the measured surface spectral irradiance and $\Delta a$ is the difference between the spectral albedo of snow in the presence and absence of impurities between the wavelengths $\lambda_1$ and $\lambda_2$. For this study, $\lambda_1$ = 350 nm (the shortest wavelength used) and $\lambda_2$ = 1000 nm, where we expect very little change due to the presence of impurities (Painter et al., 2013). Pure snow albedo is estimated through SNICAR and incorporating snowpack and sky conditions. Before BrC deposition, the natural snowpack had an instantaneous radiative forcing of 51.7 W m$^{-2}$ and 35.2 W m$^{-2}$ for the two natural snow sites at Tamarack Lake, where the spectral albedo measured included all light-absorbing impurities inherently present. Considering the difference in measured spectral albedo of the natural snowpack and after depositing BrC aerosol from the combustion of AK peat (Fig. 3), the deposition experiment added an additional 14.0 W m$^{-2}$ and 19.8 W m$^{-2}$, for a total instantaneous RF of 65.7 W m$^{-2}$ and 55.0 W m$^{-2}$ for Experiment 1 and 2, respectively. If we integrate the deposited BrC concentrations for L1, L2, and L3 for each





deposition, the result in an average mass-weighted instantaneous RF of 1.23 (+0.14/-0.11) W m$^{-2}$ per ppm of BrC (or OC) aerosol deposited from the combustion of AK peat.

### 3.2.3 Radiative forcing by impurities as a function of existing conditions

The radiative impact of the deposition of light-absorbing impurities in snow is dependent on multiple variables including snow
age (i.e., grain size), existing impurity content prior to deposition, and solar/sky conditions (Skiles et al., 2018). The rate at which the albedo is lowered, and thus the RF increases, by increasing total impurity loading decreases as the impurity concentration increases, a direct result of the great differences between the absorption properties of the impurities and that of ice itself  in the UV-visible wavelength ranges (Warren and Wiscombe, 1980).

    For example, we consider consider the rate of the direct albedo effect on RF using a SNICAR-modeled snowpack,
with a 5-cm thick surface layer and an optically semi-infinite layer below that, both with densities of 200 kg m$^{-3}$ and effective grain radii of 500 μm. In Fig. 9, we demonstrate the reduction of broadband albedo $\alpha_{BB}$ evaluated between 350 nm and 1000 nm as function of concentrations of BC in the surface layer ranging from 0 ppb to an unrealistic amount of 50 ppm As expected, $\alpha_{BB}$ is rapidly reduced from a pure snow value of ~0.90 until the resulting $\alpha_{BB}$ is completely dominated by BC and the albedo approaches reported values of the SSA of freshly generated BC (Bond et al., 2013). If we assume the sky conditions during
the deposition experiments in Sect. 3, the RF can be calculated using Eq. (13), where $\Delta\alpha$ is now the idfference between the pure snow spectral albedo and that of the snow with increasing values of BC. The RF dramatically increases with small amounts of BC added to the surface layer of a pristine snowpack, and then begins to saturate as the layer becomes optically thick with BC. Similar to $\alpha_{BB}$, the rate at which the RF increases with increasing BC loading is reduced until it is reduced to zero. These results are similar for the mineral dust component of SNICAR as well as for the BrC component, using optical properties from
this study. That is, we can apply this same concept to the deposition of BrC on pristine snow using SNICAR and the average BrC optical properties from this study. Using the snowpack and irradiance properties from the experiments described in Sect. 3, we simulate the deposition of just BrC aerosol onto a pristine snowpack. Again, as expected, the RF values are greater than those found in the previous section, where the BrC deposited onto pure snow results in RF of 25.8 W m$^{-2}$ and 53.3 W m$^{-2}$ for the first and second experiment, respectively. And, the resulting RF efficiency increases from 1.23 (+0.14/-0.11) W m$^{-2}$ per
ppm of BrC deposited to 2.68 (+0.27/-0.22) W m$^{-2}$ per ppm of BrC deposited onto a pristine snowpack. While these examples don't examine the resulting effects of the grain-size feedback (lowering albedo due to grain size growth from enhanced metamorphism from light absorption of impurities heating of the snowpack), they provide insight and quantify differences between instantaneous radiative forcing of impurities deposited onto pure snow versus onto older snow with already existing impurities.



## 4 Addressing uncertainty and sources of error

This study provided an estimation of the imaginary refractive index $\kappa$ of BrC artificially deposited on a snowpack. While the authors have taken care to limit the amount of uncertainty throughout the study, the analyses and calculations used in this study are not without assumptions and sources of error. These are addressed below.

### 4.1 Deposition experiment, spectral albedo measurement, and snow sample collection

Artificially depositing aerosols onto the snow surface is a proven method to study aerosol-cryosphere interactions (Beres and Moosmüller, 2018; Brandt et al., 2011; Conway et al., 1996; Peltoniemi et al., 2015). For this study, aerosol deposition, spectral albedo measurement, and the collection of snow samples were conducted as quickly as possible while minimizing contamination in an effort to reduce uncertainties discussed below.

In this experiment, the deposition apparatus increased the snow effective grain radii at the snow surface through localized heating due to covering the snowpack with the deposition volume during ambient conditions of warm temperatures and high solar insolation. While temperature inside the deposition volume was not monitored, this effect was observed by examining the difference in measured NIR (700 – 1300 nm) albedo, where snow reflectance is most sensitive to changes in grain size. Additionally, the presence of liquid water may also have been enhanced due to this local heating of the snow surface, and the induced melt may have contributed to impurities percolating lower into the snowpack.

Another observation made over several BrC deposition experiments is the loss of BrC absorption over time. While the spectral albedo measurements in our study were made within 5 minutes of aerosol deposition, and likely were not affected significantly by this phenomenon, unpublished data of the hemispherical-conical reflectance factor (HCRF, Schaepman-Strub et al., 2006) of the snow surface after the deposition of BrC aerosols from AK peat and another BrC aerosol-producing surrogate fuel (i.e., incense, Chakrabarty et al. (2013) and Gyawali et al., 2012) showed that the UV-reflectance increased over some time after deposition. In one instance, the HCRF at 350 nm increased from 0.69 to 0.83 in 207 minutes, whereas the HCRF of the natural snow nearby did not exhibit any discernible change over the same time period. We hypothesize that there are a couple factors that may have contributed to this increase. First, experiments were conducted during the winter/spring seasonal transition, where warm air temperatures induced melt in the surface layer(s) of the snow (the snowpack was not isothermal). Liquid water present in the snowpack flushes water-soluble OC compounds through the snowpack, which may have occurred in the top layers of the snow. Secondly, high UV solar irradiance may photobleach and heat from absorbed radiation may volatilize chromophoric organic compounds (Bertilsson and Tranvik, 2000; Grannas et al., 2007; Laskin et al., 2015; Sumlin et al., 2017). These absorption-reducing photochemical processes may also apply to BrC aerosol deposited on snow, and the resulting change in snowpack chemistry and optics are not fully understood. Likely, the reason for the reduction in absorption is a combination of above scenarios and further work is needed to quantify these effects.



### 4.2 TOC concentration measurements

The concentrations of TOC presented in this paper represent those of melted snow and its impurities. However, while care was taken to minimize the contamination of snow samples collected, a small amount of TOC was likely scavenged from the Whirlpak plastic bags used to store and transport snow samples. We have derived an upper limit of this effect by measuring the TOC concentration of UPW shaken in new Whirlpak bags to be $0.493 \pm 0.025$ g m$^{-3}$; we consider this to be the upper-limit of contamination from liquid water and we would expect this value to be much smaller as our samples remained frozen until TOC determination. Additionally for this study, we did not filter the samples prior to TOC determination, as many studies have done using 0.45 or 0.22 μm pore size filters, to determine "dissolved" organic carbon (DOC). Instead, we assume that the total OC measured represents water-soluble and insoluble OC as well as particulate and non-particulate OC. With regard to soluble versus insoluble OC, the manufacturer of the instrument used for this study states that the recovery of insoluble OC in the instrument is greater than 90%. There are no data from the manufacturer regarding particulate OC (POC) versus non-particulate OC measured using a TOC analyzer; however, Potter and Wimsatt, (2012) indirectly indicate that the mean DOC and TOC concentrations measured for seven unfortified source waters showed a nearly one-to-one relationship (TOC = 1.09·DOC; $R^2 = 0.997$) using a similar instrument as the one used in this study.

### 4.3 Absorbance and determination of $\kappa_{BrC}$

Measurement of the UV-vis absorbance is a common practice to determine the optical properties of atmospheric aerosols and impurities in rivers, lakes, oceans, and snow and ice. However, there are some uncertainties with regard to these methods.

While the authors assume that all of the measured absorption from impurities in the melted snow samples is solely due to BrC, it is likely that small fractions of BC (both from the deposition experiment as well as naturally occurring) and light-absorbing mineral dusts are present as well, and mineral dust poses a particular risk in assessing $\kappa_{BrC}$. The absorption of atmosphere mineral dusts is typically dominated by iron components (e.g., hematite, goethite), particularly in the UV and at short visible wavelengths (Engelbrecht et al., 2016; Moosmüller et al., 2009; Sokolik and Toon, 1999; Zhang et al., 2015). Skiles et al. (2017) developed a method to determine $\kappa$ for dusts found in the snow of Colorado mountains in high concentrations, but there has been no similar study performed for impurities in Sierra Nevada snow, where sources are much different. Sterle et al. (2013) found dust mass concentrations in the upper 30 cm of snow in the Eastern Sierra Nevada to be between 1-44 ppm in the 2008-2009 winter season, which is not insignificant. Here, we made no attempt to measure the concentrations of mineral dust, BC, or other inorganic material in Sierra Nevada snow.

In addition, some of the raw, high-spectral-resolution absorbance data had a poor signal-to-noise ratio; we improved this by smoothing the raw absorbance spectra through a moving average smoothing algorithm of 15 data points (corresponding to 15 nm) on either side of the data point in question.



## 5 Summary

Light-absorbing OC (or BrC) from smoldering biomass combustion present near snow and ice environments induces UV and short-visible light-absorption that has previously been unaccounted for in snow albedo, energy balance, and radiative forcing modeling. Biomass combustion of peat at high latitudes of the Northern hemisphere has particular potential of reducing snow and ice albedo due to the close proximity of snow and ice to BrC-rich emissions from peatland wildfires at high latitudes.

This study provides a first estimation of the spectral signatures of BrC particulate matter, from peat biomass combustion, present in the snowpack. Beres and Moosmüller (2018) have artificially deposited combustion aerosols directly onto and into the snowpack. They have shown that the deposition of emissions from small-scale smoldering combustion of Alaskan peat is effective in altering the snow surface reflectivity, especially in the UV and short-visible wavelength region. Here, we utilized this same method, together with UV-vis spectroscopy of melted snow samples, TOC concentrations in the snow before and after deposition, and Mie theory-calculated optical properties to estimate the imaginary refractive index of deposited BrC. This method has been shown to generally agree with other studies investigating these optical properties, and, by incorporating a range of derived values into SNICAR, modeled spectral albedos are shown to agree with measured albedos as well, within 5% across the UV-visible wavelength region. The instantaneous radiative forcing of BrC deposited onto the natural snowpack was shown to have a mass-weighted value of 1.23 (+0.14/-0.11) W m$^{-2}$ per ppm of BrC (or combustion OC) deposited, while deposition onto a clean (without other light-absorbing impurities) snowpack would have resulted in a more than twice as large instantaneous radiative forcing of 2.68 (+0.27/-0.22) W m$^{-2}$ per ppm of BrC deposited. These results can further inform the impact of deposited combustion aerosol on snow albedo and radiative forcing. In addition to the RF, BrC deposition can greatly reduce UV actinic flux, thereby reducing photochemistry in the snowpack. Further investigations are necessary to refine this method, as well as to address some uncertainty in the behavior of BrC in snow.

*Data Availability:* Data will be made available upon request by the corresponding author.

*Author Contributions.* NDB and HM designed the experiments and NDB, HM, and DS carried them out. NDB performed measurements and analysis of data. NDB and HM prepared the manuscript with contributions from DS, VS, and AYK.

*Competing Interests.* The authors declare that they have no conflicts of interest.

*Acknowledgements.* This material has been supported in part by NASA NNX14AN24A and NNX15AI48G and by NSF AGS-1544425. The authors acknowledge the support of Adam Watts for supplying the Alaskan peat samples, Michelle Matus for assisting with fieldwork during BrC depositions, and Nathan Chellman and the Ice Core Laboratory at the Desert Research Institute for help with TOC determination.



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





**Figure 1.** Overview of methods used to derive BrC optical properties and compare measured and modeled albedo.


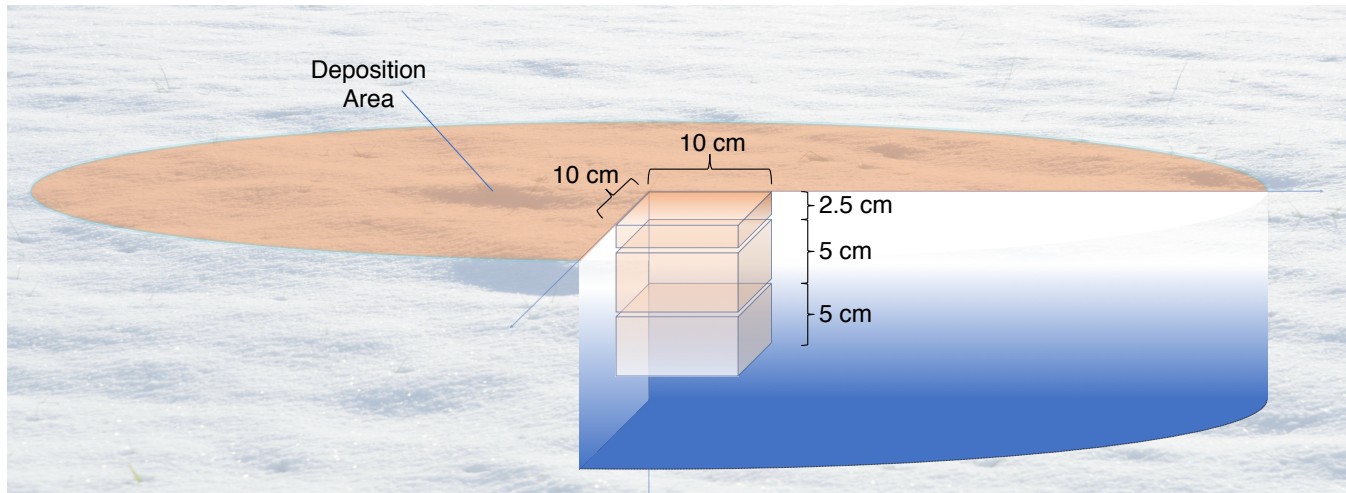

**Figure 2.** Samples for analysis are collected in three vertical layers. For the deposition site, as depicted here, samples are taken from the center of the area. Samples to represent the natural snowpack are collected similarly upwind of the deposition to minimize contamination.

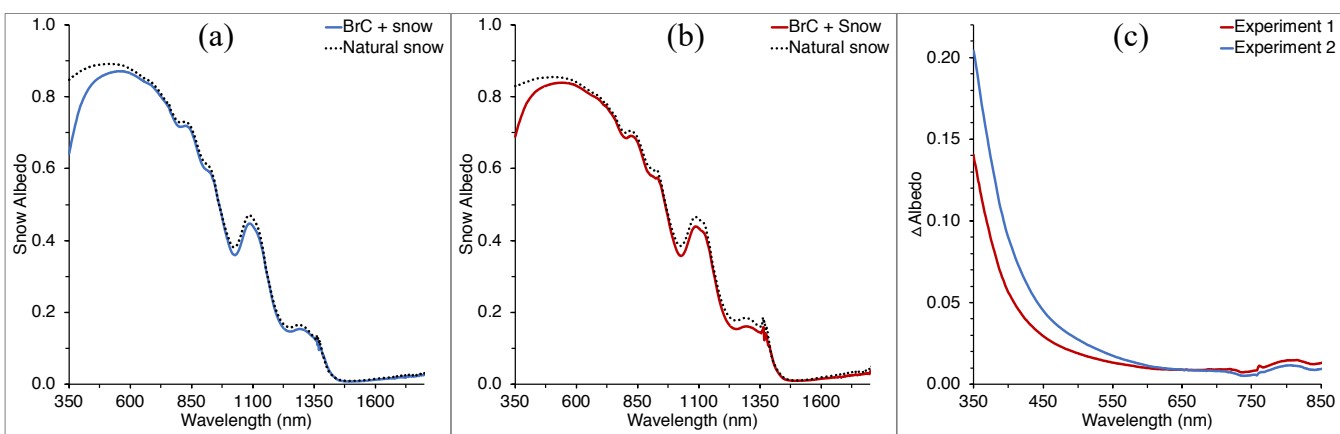

5 **Figure 3.** Panels (**a**) and (**b**) show the measured spectral snow albedo of the deposition area for Experiment 1 and Experiment 2, respectively, along with an adjacent area of natural snow. Panel (**c**) indicates the difference between the spectral albedo of the snow with and without deposited BrC in the UV-vis wavelength range, indicating a strong wavelength dependence of absorption by impurities deposited on the snow.



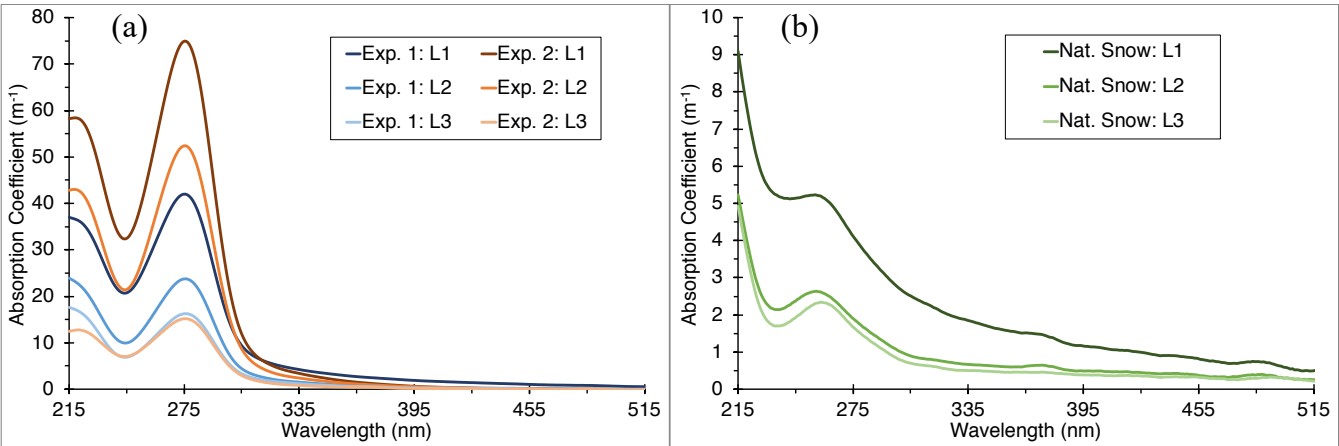

**Figure 4.** Absorption coefficient of snow meltwater for layers L1, L2, and L3 for deposited BrC (Panel **a**) and for background impurities occurring in the natural snowpack (Panel **b**). "Exp. 1" and "Exp. 2" refer to the first and second BrC deposition experiments, respectively. The spectra of the natural snow (Panel **b**) have been averaged for two locations at the experiment site.

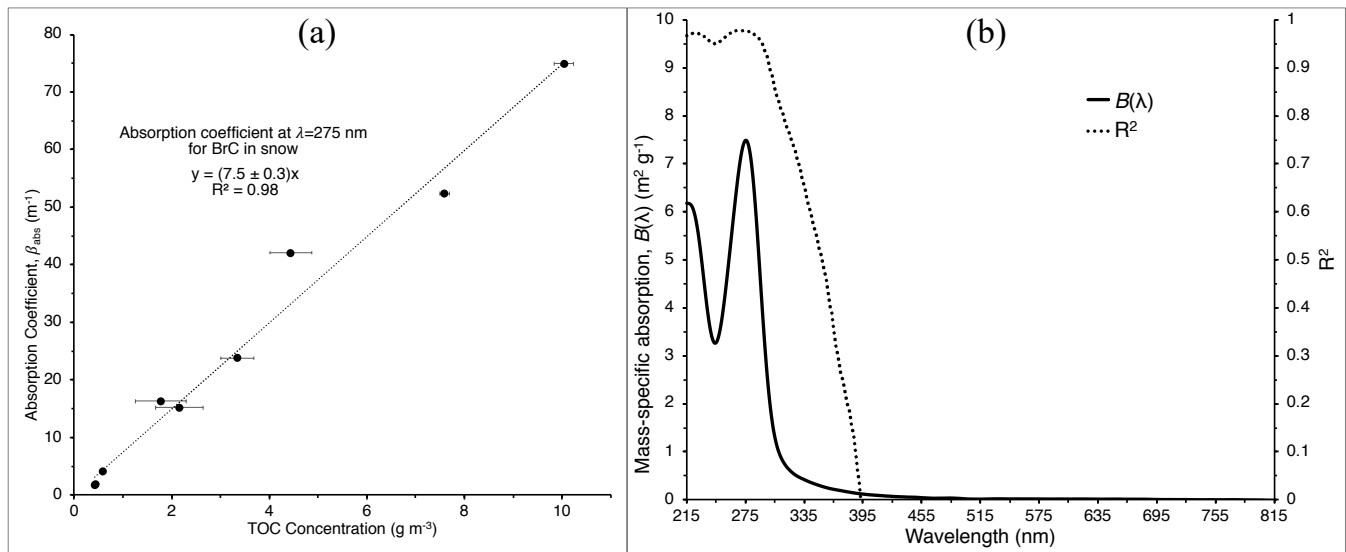

**Figure 5.** Panel (**a**): Absorption coefficient at 275 nm as function of total organic carbon (TOC) concentration. The fitted linear regression slope gives the mass-specific absorption $B_\lambda = 7.5 \pm 0.3$ m$^2$ g$^{-1}$ with a correlation coefficient of $R^2 = 0.98$. Panel (**b**): Mass-specific absorption, $B(\lambda)$, across the wavelength range $215 - 815$ nm for BrC deposited experimentally and found in natural snow at Tamarack Lake together. These panels include absorption coefficients of all meltwater samples characterized in Fig. 2.





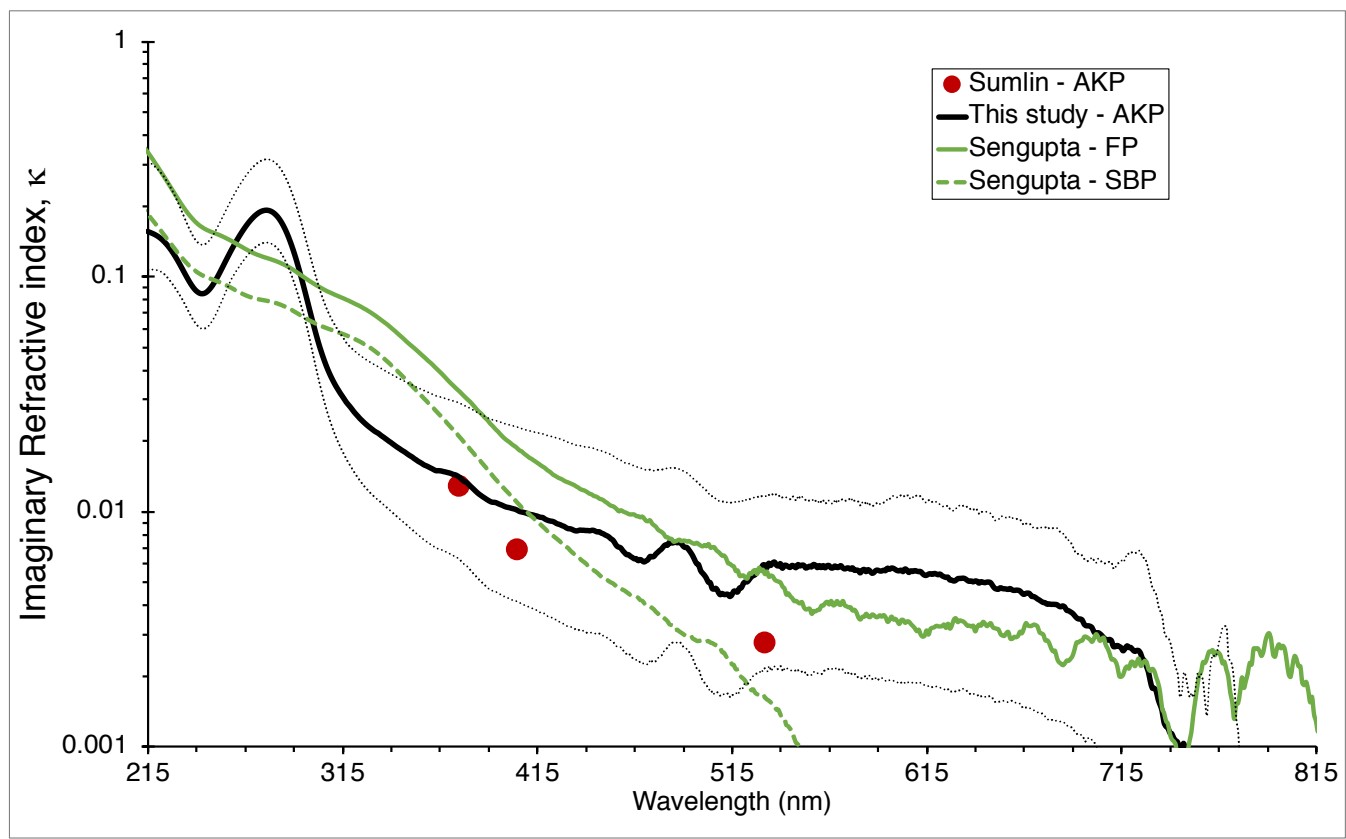

**Figure 6.** BrC imaginary refractive index, $\kappa_{BrC}$, as estimated in this study and compared to values from selected other studies: Siberian peat (unpublished data) and Florida peat – SBP and FP, respectively - from Sengupta et al. (2018), and Alaskan peat (AKP) from this study and Sumlin et al. (2018b). The thin, dotted black lines indicate the upper and lower bounds of $\kappa_{BrC}$ retrievals for this study.

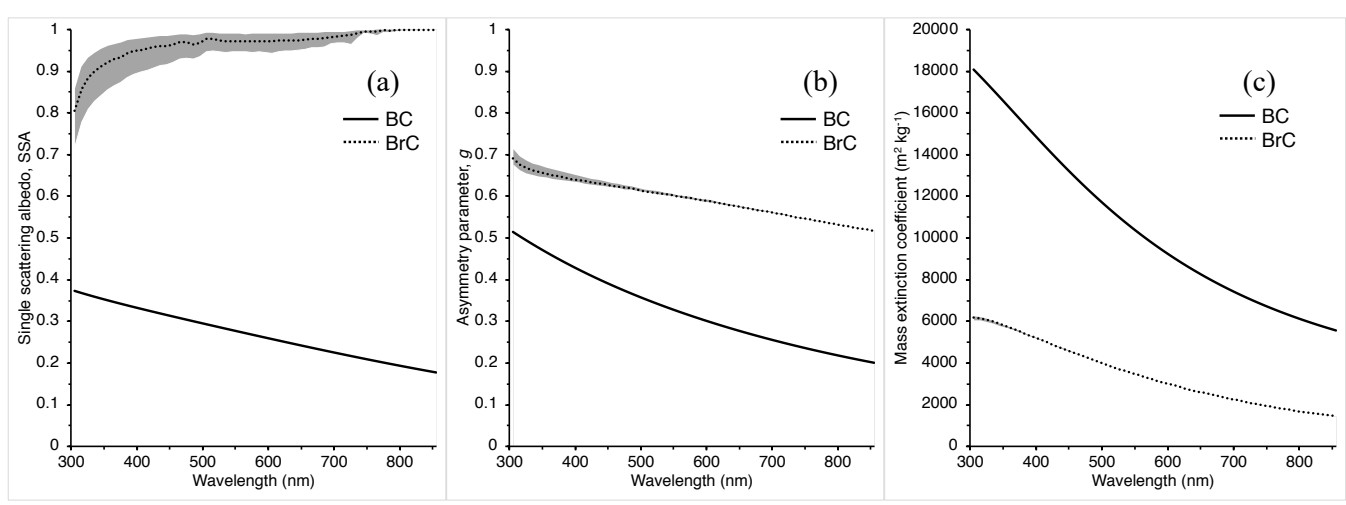

**Figure 7.** Single scattering albedo (SSA, Panel **a**), asymmetry parameter (g, Panel **b**), and mass extinction coefficient (MEC, m² kg⁻¹, Panel **c**) of BrC derived using Mie theory. BC optical properties used in the SNICAR model are provided for reference. The grey area represents the outputs of each parameter for the range of $\kappa_{BrC}$ retrievals used in the Mie code, where the largest relative discrepancy lies in the SSA.

**Figure 8.** Comparisons of measured and modeled spectral albedos and their differences for Experiment 1 (top row) and Experiment 2 (bottom row). The left column shows modeled and measured albedo of the natural snowpack at Tamarack Lake, NV. The middle column shows the measured albedo compared to SNICAR-calculated albedo by adding concentrations of deposited BrC into the model. The right column represents the difference between the measured albedo and the mean modeled albedo for the natural snow and deposited BrC. The range of modeled albedos from the range of $\kappa_{BrC}$ retrievals for BrC deposited is represented by shaded regions in the left and middle columns.



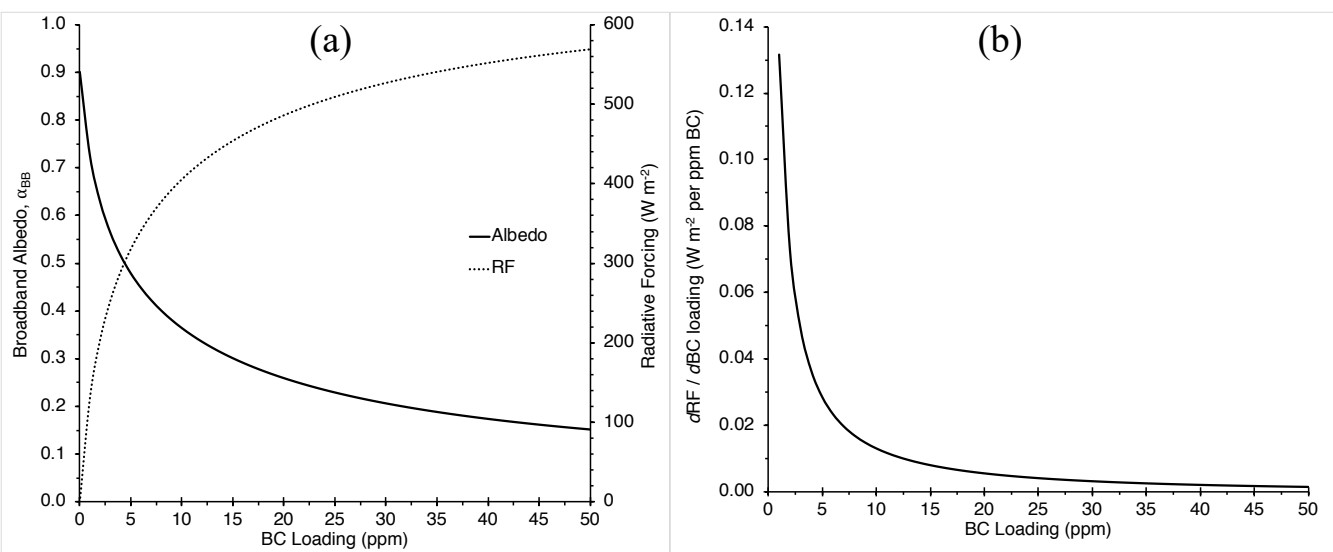

**Figure 9.** Panel **a**: The relationship between broadband albedo, $\alpha_{BB}$, and radiative forcing as a function of increasing BC concentration for a hypothetical snowpack. Panel b: Changing radiative forcing efficiency as a function of increasing BC impurity loading in the snow, demonstrating the diminishing increase in radiative forcing with ever increasing values of impurity loading.



**Table 1. Snowpack observations and spectral albedo properties**

| Site | Snow Layer (cm below surface) | Density (kg m$^{-3}$) | Surface Eff. Grain Radii[a] (μm) | SZ[b] (°) | Avg. air temp.[c] (°C) |
|---|---|---|---|---|---|
| Natural Snow 1 | 0 - 2.5 | 480 | 682 | 39.70 | 12.2 |
|  | 2.5 - 7.5 | 400 |  |  |  |
|  | 7.5 - 12.5 | 410 |  |  |  |
| Natural Snow 2 | 0 - 2.5 | 480 | 605 | 29.28 | 14.9 |
|  | 2.5 - 7.5 | 440 |  |  |  |
|  | 7.5 - 12.5 | 450 |  |  |  |
| Deposition 1 | 0 - 2.5 | 480 | 807 | 39.87 | 12.2 |
|  | 2.5 - 7.5 | 400 |  |  |  |
|  | 7.5 - 12.5 | 360 |  |  |  |
| Deposition 2 | 0 - 2.5 | 480 | 695 | 29.37 | 14.9 |
|  | 2.5 - 7.5 | 440 |  |  |  |
|  | 7.5 - 12.5 | 350 |  |  |  |

[a]Effective grain radii are estimated for the snow surface using SNICAR by matching the measured albedo at the 1030 ice absorption feature to a modeled albedo at the same wavelength (average of 1025 and 1035 nm).
[b]Solar zenith angle at the time of albedo spectra retrieval
[c]Average air temperature during albedo spectra retrieval, as measured from the Mt. Rose SNOTEL site, which lies 0.9 km southeast at 2683 m in altitude.

**Table 2. TOC concentrations for the natural snowpack and snow after deposition**

| Site | Snow Layer (cm below surface) | TOC (g m$^{-3}$)[a] | TOC Uncertainty (g m$^{-3}$)[a] |
|---|---|---|---|
| Natural Snow 1 | 0 - 2.5 | 0.593 | 0.029 |
|  | 2.5 - 7.5 | 0.388 | 0.018 |
|  | 7.5 - 12.5 | 0.433 | 0.080 |
| Natural Snow 2 | 0 - 2.5 | 0.565 | 0.010 |
|  | 2.5 - 7.5 | 0.485 | 0.055 |
|  | 7.5 - 12.5 | 0.417 | 0.014 |
| Deposition 1 | 0 - 2.5 | 5.018 | 0.412 |
|  | 2.5 - 7.5 | 3.783 | 0.286 |
|  | 7.5 - 12.5 | 2.200 | 0.519 |
| Deposition 2 | 0 - 2.5 | 10.618 | 0.188 |
|  | 2.5 - 7.5 | 8.024 | 0.051 |
|  | 7.5 - 12.5 | 2.582 | 0.482 |
| BrC[b] - Experiment 1 | 0 - 2.5 | 4.425 | 0.441 |
|  | 2.5 - 7.5 | 3.395 | 0.304 |
|  | 7.5 - 12.5 | 1.767 | 0.599 |
| BrC[b] - Experiment 2 | 0 - 2.5 | 10.053 | 0.198 |
|  | 2.5 - 7.5 | 7.540 | 0.106 |
|  | 7.5 - 12.5 | 2.164 | 0.496 |

[a] (ppb = μg L$^{-1}$ = mg m$^{-3}$)
[b]BrC TOC is the difference between TOC from the deposition and the natural snow