# Peer review of "Deposition of brown carbon onto snow: changes of snow optical and radiative properties"

_Atmospheric Chemistry and Physics, 2019_

## Referee Comment (RC1) · Anonymous Referee #1 · 24 Nov 2019

This work had a good try to detect the influence of brown carbon as a new variable artificially introduced into snow. The method was delicately designed and achieved expected results. The manuscript is well organized and is promising to be accepted at last. However, I have a big concern in the instrument analysis part.

Major The authors admitted that BC is a light-absorbing particle in snow, while how can their instrument manage only to measure TOC but avoid BC? If BC was mixingly measured, the total organic carbon could be overestimated. Section 2.2 does not introduce the instrument in detail and needs to strengthen the method's introduction, including the principle, accuracy and precision of the Sievers 900 measuring TOC.

Minor 1. Page 6 Line 19. ...(TOC) concentration and absorbance in the UV and visible wavelength ranges, respectively, at the Desert Research Institute (DRI)... 2. Page 6

[Figure]

Line 22. The organic carbon is very illusive to capture. In our previous work focusing on BC in ice, we completely excluded OC just for the same reason (Refer to 2.3 of Ming et al, 2008). Could you please present an estimation of the uncertainty, regarding the way of melting at room temperature? By the way, do you consider the newly generated bacteria inside the sample, which could, in turn, contribute some possible OC?

Ming, J., Cachier, H., Xiao, C., Qin, D., Kang, S., Hou, S., and Xu, J.: Black carbon record based on a shallow Himalayan ice core and its climatic implications, Atmos Chem Phys, 8, 1343-1352, 2008.

---

## Referee Comment (RC2) · Anonymous Referee #2 · 18 Dec 2019

The paper is aimed at studies of of changes of snow optical and radiative properties due to deposition of BrC onto snow.The authors perform the artificial deposition of BrC aerosol onto snow surfaces and monitor the spectral radiative impact of the deposited BrC. The work is sound and worth to be published in ACP. My comment is given below: The natural snow albedo is not 1.0 in the visible ( see Fig.3). Therefore, it is clear that the snow samples were already polluted before introduction of BrC. I think, the better idea would be to use fresh or artificial (not polluted) snow samples.

---

## Referee Comment (RC3) · Anonymous Referee #3 · 25 Dec 2019

This paper describes and experiment whereby smoldering combustion is used to add light-absorbing aerosol (BC, BrC) to a controlled area of ambient snowpack in the Sierra Nevada mountains. The total organic carbon in three layers and the snowpack albedo are measured for ambient snow in an adjacent area unaffected by the intentional addition of aerosol and in the affected area. Spectral absorption by soluble and insoluble components in each of the three layers in both snowpacks was measured in the lab, using a spectrophotometric measurements of transmission of light through melted snow samples. From this, bulk absorption properties of snow impurities was measured, and spectral mass absorption efficiencies (absorption per mass of total organic carbon, TOC) and imaginary index of refraction was calculated for 215-815nm wavelengths. Assuming absorption was due to spherical particles BrC (where "BrC"

mass is the TOC mass and the aerosol real index of refraction is fixed at 1.6) Mie code is used to calculate the BrC single scatter albedo, asymmetry parameter and mass extinction coefficient. These are used as inputs to SNICAR code to derive modeled snow albedo, which is compared to the measured spectral albedos. Radiative forcing is calculated in SNICAR and is used to derive forcing per mass of combustion aerosol (as measured by total organic carbon mass) for the ambient snowpack and for a pure snowpack.

The analysis presented is largely robust and appears to be based on a carefully executed experiment. Overall, I think the paper is excellent and contributes useful information on the spectral optical properties of BrC in snow – and also possibly in the atmosphere. There are a few issues I'd like to see addressed.

The paper is well-organized and well-written. Below my main comments I suggest a number of small edits.

MAIN COMMENTS:

Abstract:

I think a more useful result emerging from this work is the calculation of spectrally-resolved mass absorption efficiencies for TOC:absorption across 275-815nm. As noted below, the authors have not convinced me that this is purely a BrC MAE; in particular, BC is likely contributing to measured absorption. Nonetheless, given the paucity of observed values of spectral absorption by BrC, this dataset will be useful. As such, I think this aspect of the work should be highlighted in the Abstract.

The wording in line 23 of the Abstract needs to be modified so the focus isn't on the magnitude of the RF but the forcing per mass of deposited aerosol. I'd suggest editing to, e.g.: "The instantaneous radiative forcing per unit mass of total organic carbon deposited to the ambient snowpack was found to be 1.23 (+0.14/-0.11) W/me per ppm. This snowpack already contained light-absorbing impurities; in a completely

clean snowpack the forcing per mass TOC deposited would have been 2.68(+0.27/-0.22) W/m2 per ppm of BrC deposited."

Main paper:

1) The paper reads as if BrC will exist as an aerosol independent of other aerosol components (e.g. BC). In the ambient atmosphere BrC and BC pretty quickly become internally mixed. There are two aspect of this work that are likely not reflective of the 'real world', and which should be acknowledged: Here, the aerosol are generated in what sounds like fairly realistic peat-burning conditions (kudos to the team for this aspect of the work!). However, the aerosol are deposited to snow within minutes (or less), whereas in the real world most deposited aerosol will have had, at a minimum, hours to chemically evolve and mix. As widely documented, the chemical and optical properties of combustion aerosol – and the organics in particular – rapidly evolve in the first hour or few hours after emission. It needs to be acknowledged that real ambient aerosol that is deposited is likely internally mixed, so BrC "aerosol" doesn't really exist, whereas the aerosol deposited in this experiment is likely closer to an external mixture (e.g. pg. 3, lines 5-8 reads as if BrC "aerosol" exist in the ambient; these are likely mostly internally mixed aerosol with both BrC and BC). Second, and more important, is that ambient deposited aerosol is likely at least hours and mostly closer to days old, so the chemical composition and therefore optical properties of combustion aerosol, and the BrC component in particular, may be substantially different from the aerosol measured in the reported experiment.

2) pg. 8, lines 16-20 and It's noted that 700-860nm absorbance was subtracted from the measured signal, as a way of normalizing for drift.

a) What is the source of this "drift"? (instrumental?)

b) The authors comment that this reduces the influence of any BC particle absorption since this is "fairly independent of wavelength across the UV-vis spectrum". However, this isn't really true: As stated in the paper itself, AAE for BC is ∼1.0 – a slope of 1

in log-log space, not a slope of zero. So this is a sort of partial removal of BC signal, but not total. As the authors are trying to isolate absorption by BrC, a better approach would be to use the stated AAE of 1.0 for BC to subtract the estimated BC absorbance from the measured signal (starting with the assumption that all 700-860nm absorption is due to BC). Why not do this?

3) pg. 9, lines 27-29: "In the UV, there was a stronger reduction (in albedo) than in the visible. . . At 700nm however albedo was reduced by less than 0.01, indicating very little BC added to the snowpack through the deposition experiments". Perhaps I'm misunderstanding: Won't this by definition be the case since the absorptance spectra were effectively zero'd out at 700-860n (pg 8, lines 16-20) for all spectra?

In general, I think the paper needs to do a clearer job of convincing me that what was measured was the spectral absorption of BrC, not a mix of BrC and BC.

Similarly, "TOC" is used synonymously with "BrC", when some of the organics in fact may not be light-absorbing (and therefore not BrC, by definition).

4) Section 3.2.2 "Estimating changes in snow albedo and radiative forcing by BrC deposition" should be split into two sub-sections and renamed:

Pg. 13, line 10 through pg 14 line 17 is really describing how the observations were used to optimize the model so it produces the observed albedos.

Pg. 14 line 18 through pg 15 line 2 is really about the forcing \*efficiency\* per mass of deposited organics in a clean snowpack versus in the specific ambient snowpack where they did the experiment. Because the experiment was not producing realistic amounts of deposited aerosol (and was not intended to) the calculated "forcing" is not itself meaningful.

5) pg. 14, lines 9-10: It's highlighted that the difference in spectral albedo between the modeled and observed albedos was less than 0.15. However, isn't this a result of the fact that BC and mineral dust amounts used in the model were specifically specified to

reduce differences in the observed and modeled spectral albedos? This seems rather problematic, to then assert good agreement when it is built in by design!

6) Section 3.2.3 Radiative forcing by impurities as a function of existing conditions. As noted above, it is not at all a new finding that forcing for a given deposited mass will depend on how dirty the snowpack was to begin with. Given that many studies have calculated forcing by assuming a totally clean ambient snowpack it is worth emphasizing this point, but it is not worth an entire page and figure (Figure 9) to show this, as it seems rather tangential to the rest of the work presented. I think it would be sufficient to reference earlier studies making this point, and to simply give the difference in forcing efficiency for the ambient snowpack versus for a clean snowpack.

7) I think the reported experiment could be improved upon in two ways. I'd ask the authors to consider whether they agree, and if so perhaps include these as areas for future work in the discussion at the end of the paper. (Note that this is not intended as a criticism of the work presented, only as ideas that were evident to me in reading the paper):

First, conducting the same experiment on a snowpack that is at and remains below freezing for the duration of the experiment would simply things. The authors do a good job of addressing how the above-freezing temperatures affected snow grain evolution and transport of aerosol in the snowpack. However, this did complicate their analysis – and the presence of liquid water may have affected the loss of particles in the snowpack and to the Whirlpak bags post-sampling.

Second, and more substantial: As noted above, the optical properties of organic aerosol in particular may rapidly evolves over the hours to day (or more) after emission in the atmosphere and in the high-actinic-flux environment of the snowpack. While the nature of the experiment is such that it would be logistically prohibitive to try and make a mixing chamber that would allow for in-atmosphere aging of the emissions before deposition, it would be feasible to monitor the evolution in snowpack spectral albedo

over the hours to day+ following deposition. Because the experiment adds such a large concentration of absorbing impurities relative to the ambient snowpack impurities and relative to the amount of additional aerosol that would be deposited to the snowpack through natural processes over, e.g., a day, if the snowpack was below freezing (so aerosol is not being transported through the snowpack), the observed evolution in albedo could be largely attributed to the evolution in the properties of the deposited aerosol.

SMALLER COMMENTS:

Pg. 5, lines 10-11: What was the diameter of the cylinder used for the deposition end of the apparatus? (see comment below on addition of this to Figure 1).

pg. 5, lines 29-30: Small question: Is there a reason for selecting specifically a 25% increase in mass?

Pg. 6, lines 1-4: Over what surface area was albedo measured? Were any corrections required for, e.g., the shadow of the instrument itself? Any issues with fact that the deposition area is not infinite and you were using a cosine collector?

pg. 11, lines 13-15: is there a reference to substantiate this assertion about the UV absorption of aromatic nuclei is at ∼255nm?

9) pg 12, line 19: The AAEs given in Doherty et al. (2010) are for 450-600nm. AAE over this wavelength band might be quite different from that at the band of 330-400nm given here.

Figure 1: Overall this is a useful figure. I was, however, a bit confused by the box on the left side that simply says "Sumlin et al. (2018)". What is this supposed to represent? If it's the source of the BrC size distribution, just put "Sumlin et al. (2018)" inside that box.

Figure 2: Can you add the radius/diameter of the deposition area to this figure, please?

Figure 5 caption should state the R-squared at each wavelength (as shown in panel a for 275nm) is also given for each wavelength, and it should note at what wavelength resolution the mass-specific absorption is given.

Figure 7 caption should note that the real index of refraction is assumed to be a fixed value of 1.6 across all wavelengths. This is why both g and the mass extinction coefficient are so invariant (small gray area in panels b and c)!

Table 1: The air temp is given (footnote!) at a site 0.9km SE of the sampling site and at 2683m altitude. What is the altitude of the sampling site itself? Considerably different from 2683m? (Can you use the dry adiabatic lapse rate to estimate the sampling site temp if it wasn't measured, which would have been easy enough to do!)

SUGGESTED SMALL EDITS:

Pg. 2, line 10: "cryosphere are a growing" –> "cryosphere is a growing"

Pg. 2, lines 20-22: Dang and Hegg (2014) should also be cited here.

Pg. 3, line 3: "toward shorter wavelength" –> "towards shorter wavelengths"

Pg. 3, line 25: add a comma after "(Ervens et al., 2004)"

Pg. 4, line 9: spectral albedo of the non-BC components was not measured/derived, but they did report an assumed value of AAE=5 for the 450-600nm wavelength band.

Pg. 4, line 26: "presented in Fig. 1 to guide through the relationships" –> "presented in Fig. 1 as a guide to the relationships"

Pg. 7, line 24: "This method assumes there is little contribution to scattering to the overall extinction of light along the path." These samples surely include insoluble particles, so there was some contribution from particulate scattering to the signal. Was the possible magnitude of this effect estimated? Would this constitute a positive bias in derived absorption?

Pg. 9, line 25: "... in Fig. 3 alongside measured" –> "... in Fig. 3 alongside the measured"

Pg. 9, line 27: "In the UV, there was a stronger reduction than in the visible, up to ~0.14 and ~0-.21 at 350nm for Experiment 1 and 2, respectively (Fig. 3)" –> "In the UV, there was a stronger reduction in albedo than in the visible, of ~0.14 and ~0-.21 at 350nm for Experiment 1 and 2, respectively, relative to the natural snowpack albedo (Fig. 3)"

Pg. 10, line 23: "snow sampling sites, L1 TOC" –> "snow sampling sites, the L1 TOC"

Pg. 10, line 26: "air pollution from mining..." –> "air pollution from the mining..."

Pg. 10, line 30: "concentration than Experiment 1" –> "concentration as Experiment 1"

Pg. 10, lines 31-32: delete "...for Experiments 2 than for Experiment 1 relative to the natural snow" (it's not needed).

Pg. 11: "UV absorption band in presence of an aromatic.." –> "The UV absorption band in The presence of an aromatic.."

pg. 11, lines 29-30: I'm not sure what you're trying to say here with "...together with those of the nature snowpack, TOCsnow, as one dataset." Figure 5 only shows one dataset. ?

pg. 13, line 24: Figure 7 should be referenced at the end of the sentence at the beginning of this line.

Pg. 14, lines 9: the reference to Figure 7 should be to Figure 8.

---

## Author Comment (AC1) · 7 Jan 2020

**Reply to the review by Anonymous Referee #1 for the manuscript, "Deposition of Brown Carbon onto Snow: changes of snow optical and radiative properties" by N. D. Beres et al.**

We thank the anonymous reviewer for their thoughtful response and recommendations to improve the manuscript through clarification of the analysis, particularly of total organic carbon determination of the melted snow samples. Below, questions and comments by the reviewer are in blue and the responses by the manuscript authors are in black.

The authors admitted that BC is a light-absorbing particle in snow, while how can their instrument manage only to measure TOC but avoid BC? If BC was mixingly measured, the total organic carbon could be overestimated. Section 2.2 does not introduce the instrument in detail and needs to strengthen the method's introduction, including the principle, accuracy and precision of the Sievers 900 measuring TOC.

The authors agree with the reviewer's latter suggestion that the methods section concerning TOC determination (beginning on Page 6, Line 24) should be expanded to include more information regarding the Sievers 900 instrument, its analysis protocols, and reported measurement accuracy and precision. These have been included in the updated manuscript, beginning on page 6, line 26.

With respect to the reviewer's former suggestion – the instrument's ability to determine TOC from indirect/unintentional oxidation of BC in liquid samples during its normal operation – the instrument is not believed to be able to convert BC to measurable TOC through the photo-chemical oxidation methods utilized. BC is insoluble and chemically inert and recalcitrant. One supporting publication is Peltier et al. (2007), who employ a Sievers 800-series TOC analyzer (utilizing the same oxidation methods as the 900-series instrument used in our study) and were unable to detect elemental carbon (EC) in aqueous solutions.

In addition, the concentrations of BC found in the snow, both before the deposition and even after are much smaller than those of deposited TOC. Concentrations of BC measured in snow of the Sierra Nevada in the United States, for example, has concentrations in the 10s or low-100s of ppb (Hadley et al., 2010; Sterle, et al., 2013). Even visibly dirty snow may only contain ~100 ppb of BC (Gleason et al., 2019). Additionally, the mass-fraction of EC compared to that of organic carbon (OC) is very small for emissions from smoldering combustion of Siberian and Alaskan peat. For example, Chakrabarty et al. (2016) report the OC:EC mass ratio (based on emission factors) as 70:1 for Alaskan peat combustion under very similar combustion and fuel conditions to our experiment. Thus, the light-absorbing OC produced through smoldering combustion of peat (i.e. BrC) dominates the optical, physical, and chemical presence of carbonaceous particulate matter reported in this study.

Minor
1. Page 6 Line 19. ...(TOC) concentration and absorbance in the UV and visible wavelength ranges, respectively, at the Desert Research Institute (DRI)...

The authors feel that this suggested change – adding the word "respectively" – is incorrect and unnecessary. The UV wavelength range does not refer to only the TOC determination; similarly, the visible wavelength range does not refer specifically to the absorbance determination. While both instruments utilize UV wavelengths of light during their operation, only the spectrophotometer uses both UV and visible light to determine absorbance. These are two separate measurements presented and utilized in this work. Therefore, "respectively" should not be added to the statement to which referee #1 refers to. However, we clarified this in the manuscript to better express what was intended, namely that total organic carbon (TOC) concentration measurements and UV-visible spectroscopy were carried out [separately] at the Desert Research Institute.

2. Page 6 Line 22. The organic carbon is very illusive to capture. In our previous work focusing on BC in ice, we completely excluded OC just for the same reason (Refer to 2.3 of Ming et al, 2008). Could you please present an estimation of the uncertainty, regarding the way of melting at room temperature?

The present manuscript includes a comprehensive section (Section 4, Page 16, Line 1) discussing possible uncertainty and sources of error throughout the study presented. This includes a statement in which the TOC concentration of ultra-pure water (UPW) shaken in Whirlpak bags (the same used for snow sample collection and frozen storage) was measured and determined to be an upper limit of contamination possible; however, this could only possibly happen if the collected snow samples were melted and shaken in the Whirlpak bags prior to TOC determination. Earlier in the

manuscript (Page 7, Line 3), we also mention that the polyurethane vials used when melting the snow contribute a non-negligible amount of TOC to the overall determination of TOC in the melted snow samples. Indeed, this contamination and its uncertainty was propagated throughout the calculations involving TOC values, including the derivation of the imaginary part of the refractive index of brown carbon deposited on the snow surface.

By the way, do you consider the newly generated bacteria inside the sample, which could, in turn, contribute some possible OC?

We do not consider any bacteria, specifically, in the contribution to measured OC of the melted snow sample, before or after collection. That is to say, we do not partition the different species of OC within the "natural" snow samples; the goal of this work is to only consider the difference in *total* organic carbon before and after deposition of BrC, thereby isolating the contribution made by the deposition experience to the TOC. However, the OC determination is not exclusive to non-biological organic material and can indeed include bacteria that contribute to the TOC determination presented in the manuscript. We clarify this by adding a statement with regard to the possible OC sources present in the collected snow samples.
* * *
Chakrabarty, R. K., Gyawali, M., Yatavelli, R. L. N. N., Pandey, A., Watts, A. C., Knue, J., Chen, L.-W. A. W. A., Pattison, R. R., Tsibart, A., Samburova, V. and Moosmüller, H.: Brown carbon aerosols from burning of boreal peatlands: microphysical properties, emission factors, and implications for direct radiative forcing, Atmos. Chem. Phys., 16(5), 3033–3040, doi:10.5194/acp-16-3033-2016, 2016.

Gleason, K. E., McConnell, J. R., Arienzo, M. M., Chellman, N. and Calvin, W. M.: Four-fold increase in solar forcing on snow in western U.S. burned forests since 1999, Nat. Commun., 10(1), 2026, doi:10.1038/s41467-019-09935-y, 2019.

Hadley, O. L., Corrigan, C. E., Kirchstetter, T. W., Cliff, S. S. and Ramanathan, V.: Measured black carbon deposition on the Sierra Nevada snow pack and implication for snow pack retreat, Atmos. Chem. Phys., 10(15), 7505–7513, doi:10.5194/acp-10-7505-2010, 2010.

Peltier, R. E., Weber, R. J. and Sullivan, A. P.: Investigating a Liquid-Based Method for Online Organic Carbon Detection in Atmospheric Particles, Aerosol Sci. Technol., 41(12), 1117–1127, doi:10.1080/02786820701777465, 2007.

Sterle, K. M., McConnell, J. R., Dozier, J., Edwards, R. and Flanner, M. G.: Retention and radiative forcing of black carbon in eastern Sierra Nevada snow, Cryosphere, 7(1), 365–374, doi:10.5194/tc-7-365-2013, 2013.

---

## Author Comment (AC2) · 7 Jan 2020

**Reply to the review by Anonymous Referee #2 for the manuscript, "Deposition of Brown Carbon onto Snow: changes of snow optical and radiative properties" by N. D. Beres et al.**

The authors thank the anonymous reviewer for their comments and recommendation for publication. Below, comments by the reviewer are in blue and the responses by the manuscript authors are in black.

The natural snow albedo is not 1.0 in the visible (see Fig.3). Therefore, it is clear that the snow samples were already polluted before introduction of BrC. I think, the better idea would be to use fresh or artificial (not polluted) snow samples.

The experiments in this study were conducted using a simple and portable deposition device (Beres and Moosmüller, 2018) which can mimic real-world aerosol dry deposition processes of varying mass concentrations onto real-world surfaces, such as snow. The goal of this study was a first investigation of how brown carbon (BrC) produced through the combustion of an important fuel source can change snow optical and radiative properties; future investigations using the deposition apparatus can benefit from varying the snow conditions (high versus low snow mass density, varying grain radii, etc.). However, for our study, to isolate the influence from BrC only, the presence and influence of light-absorbing impurities in the snowpack before the deposition experiment is negated by finding the difference in values of spectral albedo, TOC concentrations, and spectrophotometric absorption, as measured before and after the deposition of BrC. This way, the BrC influence – on the optical properties, primarily – is isolated and investigated. As noted in the manuscript, it will require further research to refine our methods and determine some additional BrC-related effects to snowpack chemistry and optics. Using an artificial snowpack – such as the methods described in Hadley and Kirchstetter (2012) – may reduce some uncertainty in these methods while increasing others because natural and artificial snow differ in morphology, etc., but the macroscopic BrC-related effects and key results presented in our manuscript will likely remain intact.

The snow samples used for analysis in this study were indeed already "polluted" (that is to say, not "pure" or free from *all* impurities) prior to the artificial deposition of BrC, and the authors were aware of this fact. Concentrations of BC measured in snow of the Sierra Nevada in the United States, for example, have values in the 10s or low-100s of ppb and dust concentrations may be greater (Hadley et al., 2010; Sterle, et al., 2013). It should be noted that "pure" snow spectral albedo equal to 1.0 in the visible is never found in a natural snowpack, and light-absorbing impurities will be found even in snow of the most pristine areas of the Earth's cold regions, which will reduce the measured albedos or reflectance in the visible wavelength region (e.g. Warren and Wiscombe (1982), Warren et al. (2006), and Forrström et al. (2009)).

Beres, N. D. and Moosmüller, H.: Apparatus for dry deposition of aerosols on snow, Atmos. Meas. Tech., 11(12), 6803–6813, doi:10.5194/amt-11-6803-2018, 2018.

Forsström, S., Ström, J., Pedersen, C. A., Isaksson, E. and Gerland, S.: Elemental carbon distribution in Svalbard snow, J. Geophys. Res., 114(D19), D19112, doi:10.1029/2008JD011480, 2009.

Hadley, O. L., Corrigan, C. E., Kirchstetter, T. W., Cliff, S. S. and Ramanathan, V.: Measured black carbon deposition on the Sierra Nevada snow pack and implication for snow pack retreat, Atmos. Chem. Phys., 10(15), 7505–7513, doi:10.5194/acp-10-7505-2010, 2010.

Hadley, O. L. and Kirchstetter, T. W.: Black-carbon reduction of snow albedo, Nat. Clim. Chang., 2(6), 437–440, doi:10.1038/nclimate1433, 2012.

Sterle, K. M., McConnell, J. R., Dozier, J., Edwards, R. and Flanner, M. G.: Retention and radiative forcing of black carbon in eastern Sierra Nevada snow, Cryosphere, 7(1), 365–374, doi:10.5194/tc-7-365-2013, 2013.

Warren, S. G. and Wiscombe, W. J.: A Model for the Spectral Albedo of Snow. II: Snow Containing Atmospheric Aerosols, J. Atmos. Sci., 37(12), 2734–2745, doi:10.1175/1520-0469(1980)037<2734:AMFTSA>2.0.CO;2, 1980.

Warren, S. G., Brandt, R. E. and Grenfell, T. C.: Visible and near-ultraviolet absorption spectrum of ice from transmission of solar radiation into snow, Appl. Opt., 45(21), 5320, doi:10.1364/AO.45.005320, 2006.

---

## Author Comment (AC3) · 24 Jan 2020

**Reply to the review by Anonymous Referee #3 for the manuscript, "Deposition of Brown Carbon onto Snow: changes of snow optical and radiative properties" by N. D. Beres et al.**

We thank the anonymous reviewer for their thorough and thoughtful response and their recommendations to improve the manuscript. Below, questions and comments by the reviewer are in blue and the responses by the manuscript authors are in black.

Abstract:

I think a more useful result emerging from this work is the calculation of spectrally-resolved mass absorption efficiencies for TOC:absorption across 275-815nm. As noted below, the authors have not convinced me that this is purely a BrC MAE; in particular, BC is likely contributing to measured absorption. Nonetheless, given the paucity of observed values of spectral absorption by BrC, this dataset will be useful. As such, I think this aspect of the work should be highlighted in the Abstract.

The authors agree with the reviewer that the mass specific absorbance, $B_\lambda$, is a useful dataset for BrC-contaminated snow, though we respectfully disagree that this is "a more useful result" than the novel BrC deposition experiments and this closure study. We have included a statement in the abstract highlighting the $B_\lambda$ dataset.

The wording in line 23 of the Abstract needs to be modified so the focus isn't on the magnitude of the RF but the forcing per mass of deposited aerosol. I'd suggest editing to, e.g.: "The instantaneous radiative forcing per unit mass of total organic carbon deposited to the ambient snowpack was found to be 1.23 (+0.14/-0.11) W/me per ppm. This snowpack already contained light-absorbing impurities; in a completely lean snowpack the forcing per mass TOC deposited would have been 2.68(+0.27/- 0.22) W/m2 per ppm of BrC deposited."

The wording of the final lines of the abstract have been updated to better reflect the focus of these findings; The authors thank the reviewer for this suggestion.

Main paper:

1) The paper reads as if BrC will exist as an aerosol independent of other aerosol components (e.g. BC). In the ambient atmosphere BrC and BC pretty quickly become internally mixed. There are two aspect of this work that are likely not reflective of the 'real world', and which should be acknowledged: Here, the aerosol are generated in what sounds like fairly realistic peat-burning conditions (kudos to the team for this aspect of the work!). However, the aerosol are deposited to snow within minutes (or less), whereas in the real world most deposited aerosol will have had, at a minimum, hours to chemically evolve and mix. As widely documented, the chemical and optical properties of combustion aerosol – and the organics in particular – rapidly evolve in the first hour or few hours after emission. It needs to be acknowledged that real ambient aerosol that is deposited is likely internally mixed, so BrC "aerosol" doesn't really exist, whereas the aerosol deposited in this experiment is likely closer to an external mixture (e.g. pg. 3, lines 5-8 reads as if BrC "aerosol" exist in the ambient; these are likely mostly internally mixed aerosol with both BrC and BC). Second, and more important, is that ambient deposited aerosol is likely at least hours and mostly closer to days old, so the chemical composition and therefore optical properties of combustion aerosol, and the BrC component in particular, may be substantially different from the aerosol measured in the reported experiment.

The experiment outlined in our manuscript describes a process to isolate the optical and radiative effects of BrC aerosol on snow albedo. Previous studies (e.g. Chakrabarty et al. (2016) and Sumlin et al. (2018a, 2018b)) have shown large fractions of emitted aerosol from very similar combustion conditions as to our study are primary BrC aerosol – which is to differentiate from more chemically, optically, and physically aged or secondary organics that may absorb light. Here, we demonstrate the effect of these aerosols emitted from the combustion of a fuel that is found in regions physically adjacent to snow and ice regions that may or may not undergo long-range transport before depositing and affecting the radiative and optical properties of the snow. The exact extent and impact of depositions such as this has yet to be determined or estimated, but it is not outside the realm of possibility or reason for this investigation and future ones involving deposition studies with or without BrC aerosol.

2) pg. 8, lines 16-20 and It's noted that 700-860nm absorbance was subtracted from the measured signal, as a way of normalizing for drift.

Yes, the source of the "drift" is instrumental, by which the spectrophotometric analysis may be affected. As outlined in Hecobian et al. (2010) and Sengupta et al. (2018), we subtract the light-absorption signal measured in longer wavelengths by the spectrophotometer (where we expect no influence from the experiment). This helps to account for instrument drifts during analysis periods and in between zeroing. We have added the word "instrumental" to the manuscript to clarify this.

b) The authors comment that this reduces the influence of any BC particle absorption since this is "fairly independent of wavelength across the UV-vis spectrum". However, this isn't really true: As stated in the paper itself, AAE for BC is ~1.0 – a slope of 1 in log-log space, not a slope of zero. So this is a sort of partial removal of BC signal, but not total. As the authors are trying to isolate absorption by BrC, a better approach would be to use the stated AAE of 1.0 for BC to subtract the estimated BC absorbance from the measured signal (starting with the assumption that all 700-860nm absorption is due to BC). Why not do this?

First, the reviewer is right in stating that our statement was incorrectly worded: the imaginary refractive index of BC is wavelength independent and the BC aerosol absorption coefficient is inversely proportional to the wavelength over the visible and near-visible spectral regions (AAC ≈ 1). This has been updated in our manuscript. Our statement is correct in stating that our baseline correction "reduces" the influence of any BC present in the liquid samples, but, yes, it does not completely eliminate the influence of any BC aerosol present in the sample.

If the residual absorbance in the 700-860 nm band would be solely due to BC it would, indeed, be best to model the BC absorbance over the whole spectrum. However, the residual absorption in the 800-860 nm band is caused by both instrumental drift and by BC absorption, and for our measurements, it likely that instrumental drift is dominant because very little BC is produced by the nearly exclusive smoldering combustion of peat. Had we been able to isolate the signal from the presence of BC only, we would have taken the approach as suggested by the reviewer. However, a conservative and, in our opinion, a best approach was taken to uniformly subtract the 700-860 nm absorbance signal which reduces the influence of any BC present.

3) pg. 9, lines 27-29: "In the UV, there was a stronger reduction (in albedo) than in the visible. . . At 700nm however albedo was reduced by less than 0.01, indicating very little BC added to the snowpack through the deposition experiments". Perhaps I'm misunderstanding: Won't this by definition be the case since the absorptance spectra were effectively zero'd out at 700-860n (pg 8, lines 16-20) for all spectra?

The sentence within the manuscript to which you refer (Pg. 9, lines 27-29) speaks specifically to the measured in-situ spectral albedo, which was not "effectively zero'd out" - it was in fact the table-top spectrophotometer absorbance measurements that were adjusted to account for instrument drift. This sentence that you refer to attempts to convince the reader that because the measured spectral albedo of the deposited area differed from that of the natural snowpack by less than 0.01 at 700 nm, there must be only a small amount BC aerosol added through the combustion/deposition experiment, especially considering the large amounts of BrC aerosol deposited. For example, if we use the SNICAR model to simulate the snowpack and sky conditions for the natural snowpack for Experiment 1 and add 65 ng of BC to the top 2.5 cm of snow, the spectral albedo at 700 nm (average of 695 and 705 nm values) will be reduced from 0.832 to 0.823, a difference of ~0.01. While we don't quantify BC within our snow samples, the amount of BC deposited is dominated by the amount of BrC deposited (4,425 ng in the top 2.5 cm for Experiment 1!) and this can be seen optically through spectral albedo as well, mostly for values lower than 700 nm, where we expect the spectral response to be.

In general, I think the paper needs to do a clearer job of convincing me that what was measured was the spectral absorption of BrC, not a mix of BrC and BC.

While the authors don't deny that there may be some BC aerosol mass present, there are many studies that the emission ratios of OC to BC are very large for smoldering combustion of peat fuels. We employ the same methods as previous studies that BC and BrC can be separately detected using optical methods (e.g., Bahadur et al. (2012), Kirchstetter and Thatcher (2012), Lu et al. (2015)). Additionally, our manuscript utilizes very similar combustion methods to previous studies that investigate the optical properties of emissions from the smoldering combustion of peat (e.g. Chakrabarty et al. (2016), Sumlin et al. (2018)). For example, Chakrabarty et al. (2016) list emission factors that are

20 to 110 times larger than those of BC. In our study, we find further evidence that there was not a great influence from any BC present in the deposited material, through the use of different optical measurements like snow reflectance and absorbance measurements of melted snow samples.

Similarly, "TOC" is used synonymously with "BrC", when some of the organics in fact may not be light-absorbing (and therefore not BrC, by definition).

The reviewer is correct: the fraction of light-absorbing organic carbon versus non-light-absorbing organic carbon is not known to us for this particular fuel and combustion scenario and, for this study, we do not attempt to resolve it. However, throughout the optical measurements (snow surface albedo and spectrophotometric absorbance of melted snow samples), we consistently measure the same organic material without filtering. This way, we consistently use an "average" absorption of all organic compounds present and isolate those from the deposition experiment by using background subtraction.

4) Section 3.2.2 "Estimating changes in snow albedo and radiative forcing by BrC deposition" should be split into two sub-sections and renamed:

Pg. 13, line 10 through pg 14 line 17 is really describing how the observations were used to optimize the model so it produces the observed albedos.

Pg. 14 line 18 through pg 15 line 2 is really about the forcing *efficiency* per mass of deposited organics in a clean snowpack versus in the specific ambient snowpack where they did the experiment. Because the experiment was not producing realistic amounts of deposited aerosol (and was not intended to) the calculated "forcing" is not itself meaningful.

The authors agree that this separation of sections could help the reader understand the manuscript better. The manuscript has been updated to reflect this change.

5) pg. 14, lines 9-10: It's highlighted that the difference in spectral albedo between the modeled and observed albedos was less than 0.15. However, isn't this a result of the fact that BC and mineral dust amounts used in the model were specifically specified to reduce differences in the observed and modeled spectral albedos? This seems rather problematic, to then assert good agreement when it is built in by design!

Indeed, BC and mineral dust were added to the SNICAR natural/background snowpack in order to better match the measured "natural snowpack" albedo with a modeled one. This was done only for the "natural snowpack" before the deposited BrC was added within the model for comparison purposes.

The differences seen in the values to which you refer is when only after BrC is added to snowpack within SNICAR. In order to better test the derived BrC optical properties, we must make an assumption on the amount and type of impurities present before the deposition experiment. This is also done, of course, because BC and mineral dust mass were not measured. Similarly, the measured and modeled spectral albedo of the natural snowpack would not match in the 350 and 550 nm spectral range if light-absorbing OC (i.e. BrC) was not also added (note the "dip" in the measured albedo in this spectral range in Figure 3a and 3b for the natural snowpack). But, we didn't have to make an assumption here; we measured the BrC (inferred through the TOC) in the natural snowpack and added it to the natural snowpack impurity values.

6) Section 3.2.3 Radiative forcing by impurities as a function of existing conditions. As noted above, it is not at all a new finding that forcing for a given deposited mass will depend on how dirty the snowpack was to begin with. Given that many studies have calculated forcing by assuming a totally clean ambient snowpack it is worth emphasizing this point, but it is not worth an entire page and figure (Figure 9) to show this, as it seems rather tangential to the rest of the work presented. I think it would be sufficient to reference earlier studies making this point, and to simply give the difference in forcing efficiency for the ambient snowpack versus for a clean snowpack.

The authors agree in the reviewer's recommendation and appreciate their suggestion. We have removed this section and the corresponding figure.

7) I think the reported experiment could be improved upon in two ways. I'd ask the authors to consider whether they agree, and if so perhaps include these as areas for future work in the discussion at the end of the paper. (Note that this is not intended as a criticism of the work presented, only as ideas that were evident to me in reading the paper): First, conducting the same experiment on a snowpack that is at and remains below freezing for the duration of the experiment would simply things. The authors do a good job of addressing how the above-freezing temperatures affected snow grain evolution and transport of aerosol in the snowpack. However, this did complicate their analysis – and the presence of liquid water may have affected the loss of particles in the snowpack and to the Whirlpak bags post-sampling.

Second, and more substantial: As noted above, the optical properties of organic aerosol in particular may rapidly evolves over the hours to day (or more) after emission in the atmosphere and in the high-actinic-flux environment of the snowpack. While the nature of the experiment is such that it would be logistically prohibitive to try and make a mixing chamber that would allow for in-atmosphere aging of the emissions before deposition, it would be feasible to monitor the evolution in snowpack spectral albedo over the hours to day+ following deposition. Because the experiment adds such a large concentration of absorbing impurities relative to the ambient snowpack impurities and relative to the amount of additional aerosol that would be deposited to the snowpack through natural processes over, e.g., a day, if the snowpack was below freezing (so aerosol is not being transported through the snowpack), the observed evolution in albedo could be largely attributed to the evolution in the properties of the deposited aerosol.

The authors agree that to further improve the understanding of BrC in snow and the changes it may cause to the optical, chemical, and physical properties of snow, much more experimentation and analysis is needed. This study provides a first look at how BrC aerosol produced using a relevant fuel may directly change the optical and radiative impacts of snow and ice, much in the same way that experimental studies of artificial BC deposition (and comparing to modeled results) have lent themselves to understanding how solely BC changes the optical and radiative impacts of the cryosphere, although BC is never the only impurity found in a natural snowpack.

The evolution of BrC in snow over different time scales is addressed in Section 4, where we discuss uncertainty in the study. In particular, the authors have conducted preliminary experiments of temporal changes to the snow reflectance and the degradation of absorption after deposition of primary BrC on snow over time, similar to studies showing photooxidation contributing to the loss of absorptivity over time of BrC aerosols in the atmosphere (e.g. Sumlin et al. (2017)). These unpublished results are interesting and require further testing to isolate the causes of this optical change over time.

SMALLER COMMENTS:
Pg. 5, lines 10-11: What was the diameter of the cylinder used for the deposition end of the apparatus? (see comment below on addition of this to Figure 1).

The approximate areal extent of aerosols deposited on the snow surface has been added to the paragraph containing the brief description of the deposition apparatus (page 5, line 9). The diameter (~50 cm) has been added to Figure 2.

pg. 5, lines 29-30: Small question: Is there a reason for selecting specifically a 25% increase in mass?

Yes. We chose combustion conditions – such as the fuel moisture content – to be the same or as similar as possible to previous studies (i.e. Sumlin et al. (2018)) that have carried out similar experiments. This way, we would be able to recreate the combustion conditions, apply some derived values from other studies – such as the size distribution of BrC aerosol, and (hopefully) recreate similar properties of the aerosol emitted for our purposes.

Pg. 6, lines 1-4: Over what surface area was albedo measured? Were any corrections required for, e.g., the shadow of the instrument itself? Any issues with fact that the deposition area is not infinite and you were using a cosine collector?

To the best of the authors' knowledge and abilities, the in situ measurements of snow albedo were performed without shadowing the deposition area while minimizing the distance between the deposition area and the optical fiber; however, as the reviewer has pointed out, the cosine response of the fore optic attachment includes reflectance information from outside the deposition area, albeit with less weighting for the large radius areas beyond the deposition area. While we don't quantify this effect within the manuscript, the authors thank the reviewer for our oversight in addressing this uncertainty in our measurements. Our section on uncertainties of this study (Section 4) has been

updated to include the fact that a majority of the spectral information will be from within the deposition area and much less influence will be from the surrounding ambient snowpack.

pg. 11, lines 13-15: is there a reference to substantiate this assertion about the UV absorption of aromatic nuclei is at ~255nm?

In the updated manuscript, we have included two references to support this assertion.

9) pg 12, line 19: The AAEs given in Doherty et al. (2010) are for 450-600nm. AAE over this wavelength band might be quite different from that at the band of 330-400nm given here.

The values for $AAE_{400}^{330}$ reported in our manuscript are from Zhang et al. (2019) who, in their Fig. 6, report $AAE_{400}^{330}$ and not $AAE_{600}^{450}$. We assume Zhang et al. (2019) have adapted the Doherty et al. (2010) values to the wavelength range of 330-400 properly.

Figure 1: Overall this is a useful figure. I was, however, a bit confused by the box on the left side that simply says "Sumlin et al. (2018)". What is this supposed to represent? If it's the source of the BrC size distribution, just put "Sumlin et al. (2018)" inside that box.

The Sumlin et al. (2018) study lends two items to our manuscript, the size distribution of BrC aerosol particles used in this study as well as values of derived BrC imaginary refractive index, which we use only to compare against values derived through our methods; hence, there is a "T" split in the figure. There are two dashed lines in shown in the legend as well, indicating that the dashed box emphasizing that "Sumlin et al. (2018)" is from an outside study. The figure caption has been updated to clarify the figure and the use of outside studies.

Figure 2: Can you add the radius/diameter of the deposition area to this figure, please?

Figure 2 has been updated to include the approximate radius of the deposition area and has been scaled a bit to better represent relative lengths.

Figure 5 caption should state the R-squared at each wavelength (as shown in panel a for 275nm) is also given for each wavelength, and it should note at what wavelength resolution the mass-specific absorption is given.

Thank you for this suggestion; the figure caption has been updated.

Figure 7 caption should note that the real index of refraction is assumed to be a fixed value of 1.6 across all wavelengths. This is why both g and the mass extinction coefficient are so invariant (small gray area in panels b and c)!

Thank you for this suggestion; the figure caption has been updated.

Table 1: The air temp is given (footnote!) at a site 0.9km SE of the sampling site and at 2683m altitude. What is the altitude of the sampling site itself? Considerably different from 2683m? (Can you use the dry adiabatic lapse rate to estimate the sampling site temp if it wasn't measured, which would have been easy enough to do!)

The Tamarack Lake experiment site lies at an altitude of 2694m above sea level. The authors believe that the difference in altitude (~11 m) and linear distance (< 1 km) between the Mt. Rose SNOTEL site and the Tamarack Lake site does not represent a significant change in atmospheric or meteorological conditions; thus, the air temperatures provided are representative.

SUGGESTED SMALL EDITS:

Pg. 2, line 10: "cryosphere are a growing" –> "cryosphere is a growing"

Done.

Pg. 2, lines 20-22: Dang and Hegg (2014) should also be cited here.

Done.

Pg. 3, line 3: "toward shorter wavelength" –> "towards shorter wavelengths"

Done.

Pg. 3, line 25: add a comma after "(Ervens et al., 2004)"

Done.

Pg. 4, line 26: "presented in Fig. 1 to guide through the relationships" –> "presented in Fig. 1 as a guide to the relationships"

Done.

Pg. 7, line 24: "This method assumes there is little contribution to scattering to the overall extinction of light along the path." These samples surely include insoluble particles, so there was some contribution from particulate scattering to the signal. Was the possible magnitude of this effect estimated? Would this constitute a positive bias in derived absorption?

The authors agree that the wording of this sentence may be a bit misleading in our intentions and assumptions made during our analyses. Indeed, the contribution to the scattered signal from insoluble particles is not quantified but is assumed to be low enough and present in both the snow albedo measurements after deposition and the absorbance measurements of the melted samples. Thus, any bias that is present in the measurements derived from scattered light is (assumed to be) accounted for. Additionally, visual inspection of the melted snow samples did not show any cloudiness, which would indicate scattering by a large fraction of insoluble particles. Instead, the uncertainty pertaining to insoluble particulates is present in the TOC analysis, where some insoluble OC may be unaccounted for (addressed in Section 4).

Pg. 9, line 25: ". . . in Fig. 3 alongside measured" –> ". . . in Fig. 3 alongside the measured"

Fixed.

Pg. 9, line 27: "In the UV, there was a stronger reduction than in the visible, up to ~0.14 and ~0-.21 at 350nm for Experiment 1 and 2, respectively (Fig. 3)" –> "In the UV, there was a stronger reduction in albedo than in the visible, of ~0.14 and ~0-.21 at 350nm for Experiment 1 and 2, respectively, relative to the natural snowpack albedo (Fig. 3)"

Fixed.

Pg. 10, line 23: "snow sampling sites, L1 TOC" –> "snow sampling sites, the L1 TOC"

Added "the".

Pg. 10, line 26: "air pollution from mining. . ." –> "air pollution from the mining. . ."

Done.

Pg. 10, line 30: "concentration than Experiment 1" –> "concentration as Experiment 1"

Done.

Pg. 10, lines 31-32: delete ". . .for Experiments 2 than for Experiment 1 relative to the natural snow" (it's not needed).

Done.

Pg. 11: "UV absorption band in presence of an aromatic.." –> "The UV absorption band in The presence of an aromatic.."

Fixed.

pg. 11, lines 29-30: I'm not sure what you're trying to say here with ". . .together with those of the nature snowpack, TOCsnow, as one dataset." Figure 5 only shows one dataset. ?

Reference to "one dataset" refers to the fact that the TOC concentrations from the natural snow and TOC concentrations from the deposition area snow are used together to derive the value of the specific absorbance, $B_\lambda$, instead of deriving a value of $B_\lambda$ for each individual set of TOC concentrations: one for $TOC_{snow}$ and one for $TOC_{BrC}$. This strengthens the derivation of $B_\lambda$ over the wavelength range to include more TOC/absorbance pairs for higher concentrations.

pg. 13, line 24: Figure 7 should be referenced at the end of the sentence at the beginning of this line.

Added.

Pg. 14, lines 9: the reference to Figure 7 should be to Figure 8.

Fixed.

References:

Bahadur, R., Praveen, P. S., Xu, Y. and Ramanathan, V.: Solar absorption by elemental and brown carbon determined from spectral observations, Proc. Natl. Acad. Sci., 109(43), 17366–17371, doi:10.1073/pnas.1205910109, 2012.

Chakrabarty, R. K., Gyawali, M., Yatavelli, R. L. N. N., Pandey, A., Watts, A. C., Knue, J., Chen, L.-W. A. W. A., Pattison, R. R., Tsibart, A., Samburova, V. and Moosmüller, H.: Brown carbon aerosols from burning of boreal peatlands: microphysical properties, emission factors, and implications for direct radiative forcing, Atmos. Chem. Phys., 16(5), 3033–3040, doi:10.5194/acp-16-3033-2016, 2016.

Doherty, S. J., Warren, S. G., Grenfell, T. C., Clarke, A. D. and Brandt, R. E.: Light-absorbing impurities in Arctic snow, Atmos. Chem. Phys., 10(23), 11647–11680, doi:10.5194/acp-10-11647-2010, 2010.

Hecobian, A., Zhang, X., Zheng, M., Frank, N., Edgerton, E. S. and Weber, R. J.: Water-soluble organic aerosol material and the light-absorption characteristics of aqueous extracts measured over the Southeastern United States, Atmos. Chem. Phys., 10(13), 5965–5977, doi:10.5194/acp-10-5965-2010, 2010.

Kirchstetter, T. W. and Thatcher, T. L.: Contribution of organic carbon to wood smoke particulate matter absorption of solar radiation, Atmos. Chem. Phys., 12(14), 6067–6072, doi:10.5194/acp-12-6067-2012, 2012.

Lu, Z., Streets, D. G., Winijkul, E., Yan, F., Chen, Y., Bond, T. C., Feng, Y., Dubey, M. K., Liu, S., Pinto, J. P. and Carmichael, G. R.: Light absorption properties and radiative effects of primary organic aerosol emissions, Environ. Sci. Technol., 49(8), 4868–4877, doi:10.1021/acs.est.5b00211, 2015.

Sengupta, D., Samburova, V., Bhattarai, C., Kirillova, E., Mazzoleni, L., Iaukea-Lum, M., Watts, A., Moosmüller, H. and Khlystov, A.: Light absorption by polar and non-polar aerosol compounds from laboratory biomass combustion, Atmos. Chem. Phys., 18(15), 10849–10867, doi:10.5194/acp-18-10849-2018, 2018.

Sumlin, B. J., Pandey, A., Walker, M. J., Pattison, R. S., Williams, B. J. and Chakrabarty, R. K.: Atmospheric Photooxidation Diminishes Light Absorption by Primary Brown Carbon Aerosol from Biomass Burning, Environ. Sci. Technol. Lett., 4(12), 540–545, doi:10.1021/acs.estlett.7b00393, 2017.

Sumlin, B. J., Heinson, Y. W., Shetty, N., Pandey, A., Pattison, R. S., Baker, S., Hao, W. M. and Chakrabarty, R. K.: UV–Vis–IR spectral complex refractive indices and optical properties of brown carbon aerosol from biomass burning, J. Quant. Spectrosc. Radiat. Transf., 206, 392–398, doi:10.1016/j.jqsrt.2017.12.009, 2018.

Zhang, Y., Kang, S., Gao, T., Schmale, J., Liu, Y., Zhang, W., Guo, J., Du, W., Hu, Z., Cui, X. and Sillanpää, M.: Dissolved organic carbon in snow cover of the Chinese Altai Mountains, Central Asia: Concentrations, sources and light-absorption properties, Sci. Total Environ., 647, 1385–1397, doi:10.1016/j.scitotenv.2018.07.417, 2019

---

## Author Response (AR3)

**Reply to the review by Anonymous Referee #1 for the manuscript, "Deposition of Brown Carbon onto Snow: changes of snow optical and radiative properties" by N. D. Beres et al.**

We thank the anonymous reviewer for their thoughtful response and recommendations to improve the manuscript through clarification of the analysis, particularly of total organic carbon determination of the melted snow samples. Below, questions and comments by the reviewer are in blue and the responses by the manuscript authors are in black.

The authors admitted that BC is a light-absorbing particle in snow, while how can their instrument manage only to measure TOC but avoid BC? If BC was mixingly measured, the total organic carbon could be overestimated. Section 2.2 does not introduce the instrument in detail and needs to strengthen the method's introduction, including the principle, accuracy and precision of the Sievers 900 measuring TOC.

The authors agree with the reviewer's latter suggestion that the methods section concerning TOC determination (beginning on Page 6, Line 25 in the updated manuscript) should be expanded to include more information regarding the Sievers 900 instrument, its analysis protocols, and reported measurement accuracy and precision. These items have been included in the updated manuscript, beginning on Page 6, Line 27, which reads:

> "The instrument photo-chemically oxidizes organic compounds in a liquid sample through chemical oxidation with ammonium persulfate and reactions with hydroxyl radicals produced through the photolysis of water under UV irradiance. Within the instrument, the sample stream is split in two, where one analysis path (for IC) determines the $CO_2$ formed through interaction with the ammonium persulfate only, and the other path (for TC) determines the $CO_2$ produced through both ammonium persulfate and UV-induced oxidation (TC). Then, the $CO_2$ in the sample stream is measured through a patented conductivity detector. The instrument has a reported lower detection limit of 0.03 ppb TOC and accuracy of ±2% or ±0.5 ppb, whichever is greater."

With respect to the reviewer's former suggestion – the instrument's ability to determine TOC from indirect/unintentional oxidation of BC in liquid samples during its normal operation – the instrument is not believed to be able to convert BC to measurable TOC through the photo-chemical oxidation methods utilized. BC is insoluble and chemically inert and recalcitrant. One supporting publication is Peltier et al. (2007), who employ a Sievers 800-series TOC analyzer (utilizing the same oxidation methods as the 900-series instrument used in our study) and were unable to detect elemental carbon (EC) in aqueous solutions.

In addition, the concentrations of BC found in the snow, both before the deposition and even after, are much smaller than those of deposited TOC. Concentrations of BC measured in snow of the Sierra Nevada in the United States, for example, has concentrations in the 10s or low-100s of ppb (Hadley et al., 2010; Sterle, et al., 2013). Even visibly dirty snow may only contain ~100 ppb of BC (Gleason et al., 2019). Additionally, the mass-fraction of EC compared to that of organic carbon (OC) is very small for emissions from smoldering combustion of Siberian and Alaskan peat. For example, Chakrabarty et al. (2016) report the OC:EC mass ratio (based on emission factors) as 70:1 for Alaskan peat combustion under very similar combustion and fuel conditions to our experiment. Thus, the light-absorbing OC produced through smoldering combustion of peat (i.e. BrC) dominates the optical, physical, and chemical presence of carbonaceous particulate matter reported in this study.

We have added a sentence to the end of the paragraph describing the TOC analyzer summarizing the above, which reads (Page 7, Line 11):

> "To the best of the authors' knowledge, the Sievers TOC analyzer and its methods to oxidize organic compounds in liquid samples is unable to convert any BC present to $CO_2$, as BC is chemically inert and resistant to oxidation (Bond and Bergstrom, 2006), especially under the limited exposure that instrument subjects the sample to (~4 minutes)."

Minor
1. Page 6 Line 19. ...(TOC) concentration and absorbance in the UV and visible wavelength ranges, respectively, at the Desert Research Institute (DRI)...

The authors feel that this suggested change – adding the word "respectively" – is incorrect and unnecessary. The UV wavelength range does not refer to only the TOC determination; similarly, the visible wavelength range does not refer specifically to the absorbance determination. While both instruments utilize UV wavelengths of light during their operation, only the spectrophotometer uses both UV and visible light to determine absorbance. These are two separate measurements presented and utilized in this work. Therefore, "respectively" should not be added to the statement to which referee #1 refers to. However, we clarified this in the manuscript to better express what was intended, namely that total organic carbon (TOC) concentration measurements and UV-visible spectroscopy were carried out [separately] at the Desert Research Institute. This sentence now reads (Page 6, Lines 19-21):

> "Here, snow samples were processed to quantify total organic carbon (TOC) concentration using a total carbon analyzer and absorbance using a laboratory spectrophotometer at the Desert Research Institute (DRI) in Reno, NV, USA."

2. Page 6 Line 22. The organic carbon is very illusive to capture. In our previous work focusing on BC in ice, we completely excluded OC just for the same reason (Refer to 2.3 of Ming et al, 2008). Could you please present an estimation of the uncertainty, regarding the way of melting at room temperature?

The manuscript includes a comprehensive section (Section 4, Page 15) discussing possible uncertainty and sources of error throughout the study presented. This includes a statement in which the TOC concentration of ultra-pure water (UPW) shaken in Whirlpak bags (the same used for snow sample collection and frozen storage) was measured and determined to be an upper limit of contamination possible; however, this could only possibly happen if the collected snow samples were melted and shaken in the Whirlpak bags prior to TOC determination. Earlier in the updated manuscript, we also mention that the polyurethane vials used when melting the snow contributed a non-negligible amount of TOC to the overall determination of TOC in the melted snow samples (Page 7, Line 10):

> "Contamination of the polypropylene vials with TOC was measured to be $0.020 \pm 0.001$ g m$^{-3}$, which was subtracted from each of the meltwater TOC measurements."

This contamination and its uncertainty was propagated throughout the calculations involving TOC values in this study, including the derivation of the imaginary part of the refractive index of brown carbon deposited on the snow surface.

By the way, do you consider the newly generated bacteria inside the sample, which could, in turn, contribute some possible OC?

We do not consider any bacteria, specifically, in the contribution to measured OC of the melted snow sample, before or after collection. That is to say, we do not partition the different species of OC within the "natural" snow samples; the goal of this work is to only consider the difference in *total* organic carbon before and after deposition of BrC, thereby isolating the contribution made by the deposition experiments to the TOC.
* * *
Chakrabarty, R. K., Gyawali, M., Yatavelli, R. L. N. N., Pandey, A., Watts, A. C., Knue, J., Chen, L.-W. A. W. A., Pattison, R. R., Tsibart, A., Samburova, V. and Moosmüller, H.: Brown carbon aerosols from burning of boreal peatlands: microphysical properties, emission factors, and implications for direct radiative forcing, Atmos. Chem. Phys., 16(5), 3033–3040, doi:10.5194/acp-16-3033-2016, 2016.

Gleason, K. E., McConnell, J. R., Arienzo, M. M., Chellman, N. and Calvin, W. M.: Four-fold increase in solar forcing on snow in western U.S. burned forests since 1999, Nat. Commun., 10(1), 2026, doi:10.1038/s41467-019-09935-y, 2019.

Hadley, O. L., Corrigan, C. E., Kirchstetter, T. W., Cliff, S. S. and Ramanathan, V.: Measured black carbon deposition on the Sierra Nevada snow pack and implication for snow pack retreat, Atmos. Chem. Phys., 10(15), 7505–7513, doi:10.5194/acp-10-7505-2010, 2010.

Peltier, R. E., Weber, R. J. and Sullivan, A. P.: Investigating a Liquid-Based Method for Online Organic Carbon Detection in Atmospheric Particles, Aerosol Sci. Technol., 41(12), 1117–1127, doi:10.1080/02786820701777465, 2007.

Sterle, K. M., McConnell, J. R., Dozier, J., Edwards, R. and Flanner, M. G.: Retention and radiative forcing of black carbon in eastern Sierra Nevada snow, Cryosphere, 7(1), 365–374, doi:10.5194/tc-7-365-2013, 2013.

**Reply to the review by Anonymous Referee #2 for the manuscript, "Deposition of Brown Carbon onto Snow: changes of snow optical and radiative properties" by N. D. Beres et al.**

The authors thank the anonymous reviewer for their comments and recommendation for publication. Below, comments by the reviewer are in blue and the responses by the manuscript authors are in black.

The natural snow albedo is not 1.0 in the visible (see Fig.3). Therefore, it is clear that the snow samples were already polluted before introduction of BrC. I think, the better idea would be to use fresh or artificial (not polluted) snow samples.

The experiments in this study were conducted using a simple and portable deposition device (Beres and Moosmüller, 2018) which can mimic real-world aerosol dry deposition processes of varying mass concentrations onto real-world surfaces, such as snow. The goal of this study was a first investigation of how brown carbon (BrC) produced through the combustion of an important fuel source, can change snow optical and radiative properties; future investigations using the deposition apparatus can benefit from varying the snow conditions (high versus low snow mass density, varying grain radii, etc.). However, for our study, to isolate the influence from BrC only, the presence and influence of light-absorbing impurities in the snowpack before the deposition experiment is negated by finding the difference in values of spectral albedo, TOC concentrations, and spectrophotometric absorption, as measured before and after the deposition of BrC. This way, the BrC influence – on the snow optical properties, primarily – is isolated and investigated. As noted in the manuscript, it will require further research to refine our methods and determine some additional BrC-related effects to snowpack chemistry and optics. Using an artificial snowpack – such as the methods described in Hadley and Kirchstetter (2012) – may reduce some uncertainty in these methods while increasing others because natural and artificial snow differ in morphology, etc., but the macroscopic BrC-related effects and key results presented in our manuscript will likely remain intact.

The snow samples used for analysis in this study were indeed already "polluted" (that is to say, not "pure" or free from *all* impurities) prior to the artificial deposition of BrC, and the authors were aware of this fact. Concentrations of BC measured in snow of the Sierra Nevada in the United States, for example, have values in the 10s or low-100s of ppb and dust concentrations may be greater (Hadley et al., 2010; Sterle, et al., 2013). It should be noted that "pure" snow spectral albedo equal to 1.0 in the visible is never found in a natural snowpack, and light-absorbing impurities will be found even in snow of the most pristine areas of the Earth's cold regions, which will reduce the measured albedos or reflectance in the visible wavelength region (e.g. Warren and Wiscombe (1982), Warren et al. (2006), and Forrström et al. (2009)).
* * *
Beres, N. D. and Moosmüller, H.: Apparatus for dry deposition of aerosols on snow, Atmos. Meas. Tech., 11(12), 6803–6813, doi:10.5194/amt-11-6803-2018, 2018.

Forsström, S., Ström, J., Pedersen, C. A., Isaksson, E. and Gerland, S.: Elemental carbon distribution in Svalbard snow, J. Geophys. Res., 114(D19), D19112, doi:10.1029/2008JD011480, 2009.

Hadley, O. L., Corrigan, C. E., Kirchstetter, T. W., Cliff, S. S. and Ramanathan, V.: Measured black carbon deposition on the Sierra Nevada snow pack and implication for snow pack retreat, Atmos. Chem. Phys., 10(15), 7505–7513, doi:10.5194/acp-10-7505-2010, 2010.

Hadley, O. L. and Kirchstetter, T. W.: Black-carbon reduction of snow albedo, Nat. Clim. Chang., 2(6), 437–440, doi:10.1038/nclimate1433, 2012.

Sterle, K. M., McConnell, J. R., Dozier, J., Edwards, R. and Flanner, M. G.: Retention and radiative forcing of black carbon in eastern Sierra Nevada snow, Cryosphere, 7(1), 365–374, doi:10.5194/tc-7-365-2013, 2013.

Warren, S. G. and Wiscombe, W. J.: A Model for the Spectral Albedo of Snow. II: Snow Containing Atmospheric Aerosols, J. Atmos. Sci., 37(12), 2734–2745, doi:10.1175/1520-0469(1980)037<2734:AMFTSA>2.0.CO;2, 1980.

Warren, S. G., Brandt, R. E. and Grenfell, T. C.: Visible and near-ultraviolet absorption spectrum of ice from transmission of solar radiation into snow, Appl. Opt., 45(21), 5320, doi:10.1364/AO.45.005320, 2006.

**Reply to the review by Anonymous Referee #3 for the manuscript, "Deposition of Brown Carbon onto Snow: changes of snow optical and radiative properties" by N. D. Beres et al.**

We thank the anonymous reviewer for their thorough and thoughtful response and their recommendations to improve the manuscript. Below, questions and comments by the reviewer are in blue and the responses by the manuscript authors are in black.

Abstract:

I think a more useful result emerging from this work is the calculation of spectrally-resolved mass absorption efficiencies for TOC:absorption across 275-815nm. As noted below, the authors have not convinced me that this is purely a BrC MAE; in particular, BC is likely contributing to measured absorption. Nonetheless, given the paucity of observed values of spectral absorption by BrC, this dataset will be useful. As such, I think this aspect of the work should be highlighted in the Abstract.

The authors agree with the reviewer that the mass specific absorbance, $B_\lambda$, is a useful dataset for BrC-contaminated snow, though we respectfully disagree that this is "a more useful result" than the novel BrC deposition experiments and this closure study. We have included the following statement in the abstract highlighting the $B_\lambda$ dataset on Page 1, Line 20:

> "These measurements were used to first derive a BrC (mass) specific absorption ($m^2$ $g^{-1}$) across the UV-vis spectral range."

The wording in line 23 of the Abstract needs to be modified so the focus isn't on the magnitude of the RF but the forcing per mass of deposited aerosol. I'd suggest editing to, e.g.: "The instantaneous radiative forcing per unit mass of total organic carbon deposited to the ambient snowpack was found to be 1.23 (+0.14/-0.11) W/me per ppm. This snowpack already contained light-absorbing impurities; in a completely lean snowpack the forcing per mass TOC deposited would have been 2.68(+0.27/- 0.22) W/m2 per ppm of BrC deposited."

The authors thank the reviewer for this suggestion. The wording of the final lines of the abstract (Page 1, Lines 24-27) have been updated to better reflect the focus of these findings to read:

> "The instantaneous radiative forcing per unit mass of total organic carbon deposited to the ambient snowpack was found to be 1.23 (+0.14/-0.11) W m-2 per ppm. We estimate the same deposition onto a pure snowpack without light-absorbing impurities would have resulted in an instantaneous radiative forcing per unit mass of 2.68 (+0.27/-0.22) W m-2 per ppm of BrC deposited."

Main paper:

1) The paper reads as if BrC will exist as an aerosol independent of other aerosol components (e.g. BC). In the ambient atmosphere BrC and BC pretty quickly become internally mixed. There are two aspect of this work that are likely not reflective of the 'real world', and which should be acknowledged: Here, the aerosol are generated in what sounds like fairly realistic peat-burning conditions (kudos to the team for this aspect of the work!). However, the aerosol are deposited to snow within minutes (or less), whereas in the real world most deposited aerosol will have had, at a minimum, hours to chemically evolve and mix. As widely documented, the chemical and optical properties of combustion aerosol – and the organics in particular – rapidly evolve in the first hour or few hours after emission. It needs to be acknowledged that real ambient aerosol that is deposited is likely internally mixed, so BrC "aerosol" doesn't really exist, whereas the aerosol deposited in this experiment is likely closer to an external mixture (e.g. pg. 3, lines 5-8 reads as if BrC "aerosol" exist in the ambient; these are likely mostly internally mixed aerosol with both BrC and BC). Second, and more important, is that ambient deposited aerosol is likely at least hours and mostly closer to days old, so the chemical composition and therefore optical properties of combustion aerosol, and the BrC component in particular, may be substantially different from the aerosol measured in the reported experiment.

The experiment outlined in our manuscript describes a process to isolate the optical and radiative effects of BrC aerosol on snow albedo. Previous studies (e.g. Chakrabarty et al. (2016) and Sumlin et al. (2018a, 2018b)) have shown large fractions of emitted aerosol from very similar combustion conditions as to our study are primary BrC aerosol – which

is to differentiate from more chemically, optically, and physically aged or secondary organics that may absorb light. Here, we demonstrate the effect of these aerosols emitted from the combustion of a fuel that is found in regions physically adjacent to snow and ice regions that may or may not undergo long-range transport before depositing and affecting the radiative and optical properties of the snow. The exact extent and impact of depositions such as this has yet to be determined or estimated, but it is not outside the realm of possibility or reason for this investigation and future ones involving deposition studies with or without BrC aerosol.

2) pg. 8, lines 16-20 and It's noted that 700-860nm absorbance was subtracted from the measured signal, as a way of normalizing for drift.
a) What is the source of this "drift"? (instrumental?)

Yes, the source of the "drift" is instrumental, by which the spectrophotometric analysis may be affected. As outlined in Hecobian et al. (2010) and Sengupta et al. (2018), we subtract the light-absorption signal measured in longer wavelengths by the spectrophotometer (where we expect no influence from the experiment). This helps to account for instrument drifts during analysis periods and in between zeroing. We have added the word "instrumental" to the manuscript to clarify this on Page 8, Line 28 and now reads as:

> "The raw absorbance measurements were baseline-corrected to account for instrumental drift by subtracting the scan average over the 700 – 860 nm wavelength range from individual wavelength absorbance values, similar to the method outlined by Sengupta et al., (2018)."

b) The authors comment that this reduces the influence of any BC particle absorption since this is "fairly independent of wavelength across the UV-vis spectrum". However, this isn't really true: As stated in the paper itself, AAE for BC is ~1.0 – a slope of 1 in log-log space, not a slope of zero. So this is a sort of partial removal of BC signal, but not total. As the authors are trying to isolate absorption by BrC, a better approach would be to use the stated AAE of 1.0 for BC to subtract the estimated BC absorbance from the measured signal (starting with the assumption that all 700-860nm absorption is due to BC). Why not do this?

First, the reviewer is right in stating that our statement was incorrectly worded: the imaginary refractive index of BC is wavelength independent and the BC aerosol absorption coefficient is inversely proportional to the wavelength over the visible and near-visible spectral regions (AAE $\approx$ 1). This has been updated in our manuscript on Page 8, Line 30 to read:

> "In addition, this reduces some influence of BC particle light absorption across the UV-vis spectrum, while only minimally affecting BrC particle absorption that is much greater at blue and near-UV wavelengths (Bahadur et al., 2012; Chakrabarty et al., 2010; Kirchstetter and Thatcher, 2012; Lu et al., 2015; Sumlin et al., 2018b)".

Our statement is correct in stating that our baseline correction "reduces" the influence of any BC present in the liquid samples, but, yes, it may not completely eliminate the influence of any BC aerosol present in the sample.

If the residual absorbance in the 700-860 nm band would be solely due to BC it would, indeed, be best to model the BC absorbance over the whole spectrum. However, the residual absorption in the 800-860 nm band is caused by both instrumental drift and by BC absorption, and for our measurements, it likely that instrumental drift is dominant because very little BC is produced by the nearly exclusive smoldering combustion of peat. Had we been able to isolate the signal from the presence of BC only, we would have taken the approach as suggested by the reviewer. However, a conservative and, in our opinion, best approach was taken to uniformly subtract the 700-860 nm absorbance signal which reduces the influence of any BC present.

3) pg. 9, lines 27-29: "In the UV, there was a stronger reduction (in albedo) than in the visible. . . At 700nm however albedo was reduced by less than 0.01, indicating very little BC added to the snowpack through the deposition experiments". Perhaps I'm misunderstanding: Won't this by definition be the case since the absorptance spectra were effectively zero'd out at 700-860n (pg 8, lines 16-20) for all spectra?

The sentence within the manuscript to which you refer (Pg. 9, lines 27-29) speaks specifically to the measured in-situ spectral albedo, which was not "effectively zero'd out"; it was in fact the table-top spectrophotometer absorbance

measurements that were adjusted to account for instrument drift (see point 2b above). This sentence that you refer to attempts to convince the reader that because the measured spectral albedo of the deposited area differed from that of the natural snowpack by less than 0.01 at 700 nm, there must be only a small amount BC aerosol added through the combustion/deposition experiment, especially considering the large amounts of BrC aerosol deposited. For example, if we use the SNICAR model to simulate the snowpack and sky conditions for the natural snowpack for Experiment 1 and add 65 ng of BC to the top 2.5 cm of snow, the spectral albedo at 700 nm (average of 695 and 705 nm values) will be reduced from 0.832 to 0.823, a difference of ~0.01. While we don't quantify BC within our snow samples, the amount of BC deposited is dominated by the amount of BrC deposited (4,425 ng in the top 2.5 cm for Experiment 1!) and this can be seen optically through spectral albedo as well, mostly for values lower than 700 nm, where we expect the spectral response to be.

In general, I think the paper needs to do a clearer job of convincing me that what was measured was the spectral absorption of BrC, not a mix of BrC and BC.

While the authors don't deny that there may be very small amounts of BC aerosol mass present, there are many studies that the emission ratios of OC to BC are very large for smoldering combustion of peat fuels. We employ the same methods as previous studies that BC and BrC can be separately detected using optical methods (e.g., Bahadur et al. (2012), Kirchstetter and Thatcher (2012), Lu et al. (2015)). Additionally, our manuscript utilizes very similar combustion methods to previous studies that investigate the optical properties of emissions from the smoldering combustion of peat (e.g. Chakrabarty et al. (2016), Sumlin et al. (2018)). For example, Chakrabarty et al. (2016) list OC emission factors that are 20 to 110 times larger than those of BC. In our study, we find further evidence that there was not a great influence from any BC present in the deposited material, through the use of different optical measurements like snow reflectance and absorbance measurements of melted snow samples.

Similarly, "TOC" is used synonymously with "BrC", when some of the organics in fact may not be light-absorbing (and therefore not BrC, by definition).

The reviewer is correct: the fraction of light-absorbing organic carbon versus non-light-absorbing organic carbon is not known to us for this particular fuel and combustion scenario and, for this study, we do not attempt to resolve it. However, throughout the optical measurements (snow surface albedo and spectrophotometric absorbance of melted snow samples), we consistently measure the same organic material without filtering. This way, we consistently use an "average" absorption of all organic compounds present and isolate those from the deposition experiment by using background subtraction.

4) Section 3.2.2 "Estimating changes in snow albedo and radiative forcing by BrC deposition" should be split into two sub-sections and renamed:

Pg. 13, line 10 through pg 14 line 17 is really describing how the observations were used to optimize the model so it produces the observed albedos.

Pg. 14 line 18 through pg 15 line 2 is really about the forcing *efficiency* per mass of deposited organics in a clean snowpack versus in the specific ambient snowpack where they did the experiment. Because the experiment was not producing realistic amounts of deposited aerosol (and was not intended to) the calculated "forcing" is not itself meaningful.

The authors agree that this separation of sections could help the reader understand the manuscript better. The manuscript has been updated to reflect this change. Section 3.2.2, beginning on Page 13, Line 19, is now named "Utilizing models for comparison with observations". Following the reviewer's point (6) below, some content of Section 3.2.3 has been removed; the remaining content – along with the content as recommended here (Pg. 14 line 18 through pg 15 line 2) - has been added to a new Section 2.3.2, which is named "Estimating the radiative forcing by BrC deposition", which begins on Page 15, Line 1.

5) pg. 14, lines 9-10: It's highlighted that the difference in spectral albedo between the modeled and observed albedos was less than 0.15. However, isn't this a result of the fact that BC and mineral dust amounts used in the model were specifically specified to reduce differences in the observed and modeled spectral albedos? This seems rather problematic, to then assert good agreement when it is built in by design!

Indeed, BC and mineral dust were added to the SNICAR natural/background snowpack in order to better match the measured "natural snowpack" albedo with a modeled one. This was done only for the "natural snowpack" before the deposited BrC was added within the model for comparison purposes.

The differences seen in the values to which you refer is when only after BrC is added to snowpack within SNICAR. In order to better test the derived BrC optical properties, we must make an assumption on the amount and type of impurities present before the deposition experiment. This is also done, of course, because BC and mineral dust mass were not measured. Similarly, the measured and modeled spectral albedo of the natural snowpack would not match in the 350 and 550 nm spectral range if light-absorbing OC (i.e. BrC) was not also added (note the "dip" in the measured albedo in this spectral range in Figure 3a and 3b for the natural snowpack). But, we didn't have to make an assumption here; we measured the BrC (inferred through the TOC) in the natural snowpack and added it to the natural snowpack impurity values.

6) Section 3.2.3 Radiative forcing by impurities as a function of existing conditions. As noted above, it is not at all a new finding that forcing for a given deposited mass will depend on how dirty the snowpack was to begin with. Given that many studies have calculated forcing by assuming a totally clean ambient snowpack it is worth emphasizing this point, but it is not worth an entire page and figure (Figure 9) to show this, as it seems rather tangential to the rest of the work presented. I think it would be sufficient to reference earlier studies making this point, and to simply give the difference in forcing efficiency for the ambient snowpack versus for a clean snowpack.

The authors agree in the reviewer's recommendation and appreciate their suggestion. We have removed the suggested content in this section and the corresponding figure. The remaining content has been added to a new section titled "Estimating the radiative forcing by BrC deposition" beginning on Page 15, Line 1.

7) I think the reported experiment could be improved upon in two ways. I'd ask the authors to consider whether they agree, and if so perhaps include these as areas for future work in the discussion at the end of the paper. (Note that this is not intended as a criticism of the work presented, only as ideas that were evident to me in reading the paper):
First, conducting the same experiment on a snowpack that is at and remains below freezing for the duration of the experiment would simply things. The authors do a good job of addressing how the above-freezing temperatures affected snow grain evolution and transport of aerosol in the snowpack. However, this did complicate their analysis – and the presence of liquid water may have affected the loss of particles in the snowpack and to the Whirlpak bags post-sampling.

Second, and more substantial: As noted above, the optical properties of organic aerosol in particular may rapidly evolves over the hours to day (or more) after emission in the atmosphere and in the high-actinic-flux environment of the snowpack. While the nature of the experiment is such that it would be logistically prohibitive to try and make a mixing chamber that would allow for in-atmosphere aging of the emissions before deposition, it would be feasible to monitor the evolution in snowpack spectral albedo over the hours to day+ following deposition. Because the experiment adds such a large concentration of absorbing impurities relative to the ambient snowpack impurities and relative to the amount of additional aerosol that would be deposited to the snowpack through natural processes over, e.g., a day, if the snowpack was below freezing (so aerosol is not being transported through the snowpack), the observed evolution in albedo could be largely attributed to the evolution in the properties of the deposited aerosol.

The authors agree that to further improve the understanding of BrC in snow and the changes it may cause to the optical, chemical, and physical properties of snow, much more experimentation and analysis is needed. This study provides a first look at how BrC aerosol produced using a relevant fuel may directly change the optical and radiative impacts of snow and ice, much in the same way that experimental studies of artificial BC deposition (and comparing to modeled results) have lent themselves to understanding how solely BC changes the optical and radiative impacts of the cryosphere, although BC is never the only impurity found in a natural snowpack.

The evolution of BrC in snow over different time scales is addressed in Section 4 (Page 15, Line 26), where we discuss uncertainty in the study. In particular, the authors have conducted preliminary experiments of temporal changes to the snow reflectance and the degradation of absorption after deposition of primary BrC on snow over time, similar to studies showing photooxidation contributing to the loss of absorptivity over time of BrC aerosols in the atmosphere

(e.g., Sumlin et al. (2017)). These unpublished results are interesting and require further testing to isolate the causes of this optical change over time.

SMALLER COMMENTS:
Pg. 5, lines 10-11: What was the diameter of the cylinder used for the deposition end of the apparatus? (see comment below on addition of this to Figure 1).

The approximate areal extent of aerosols deposited on the snow surface has been added to the end of the paragraph containing a brief description of the deposition apparatus (Page 5, Line 20), which now reads:

> "The areal extent of deposited aerosol is approximately 0.20 m$^2$."

The radius (~25 cm) has been added to Figure 2, as seen here:

[Figure]

Yes. We chose combustion conditions – such as the fuel moisture content – to be the same or as similar as possible to previous studies (i.e., Sumlin et al. (2018)) that have carried out similar experiments. This way, we would be able to recreate the combustion conditions, apply some derived values from other studies – such as the size distribution of BrC aerosol, and (hopefully) recreate similar properties of the aerosol emitted for our purposes.

Pg. 6, lines 1-4: Over what surface area was albedo measured? Were any corrections required for, e.g., the shadow of the instrument itself? Any issues with fact that the deposition area is not infinite and you were using a cosine collector?

To the best of the authors' knowledge and abilities, the in situ measurements of snow albedo were performed without shadowing the deposition area while minimizing the distance between the deposition area and the optical fiber; however, as the reviewer has pointed out, the cosine response of the fore optic attachment includes reflectance information from outside the deposition area, albeit with less weighting for the large radius areas beyond the deposition area. While we don't quantify this effect within the manuscript, the authors thank the reviewer for pointing out our oversight in addressing this uncertainty in our measurements. Our section on uncertainties of this study (Section 4.1) has been updated to include the fact that a majority of the spectral information will be from within the deposition area and much less influence will be from the surrounding ambient snowpack; specifically to this reviewer question, we've added the following sentences, beginning on Page 16, Line 6:

> "One limitation of the deposition apparatus and measuring the hemispherical spectral albedo of the subsequent deposited aerosols is the limited areal extent of deposited material when compared to the viewing angle of the cosine-corrected receptor. Because the deposition area is not infinite, there will be a fraction of reflected light from the surrounding ambient snowpack that will influence the

measured spectral albedo; however, this fraction of information is reduced for greater angles. Great care was taken during the experiment to minimize the effect from both instrument shadowing while decreasing the distance between the optical detector and the snow surface. As such, a majority of the reflected signal is from within the deposition area and some information is from the surrounding natural snowpack."

pg. 11, lines 13-15: is there a reference to substantiate this assertion about the UV absorption of aromatic nuclei is at ~255nm?

In the updated manuscript, on Page 11, Line 24, we have included two references to support this assertion, namely Pretsch et al. (2000) and Samburova et al. (2016). The updated statement now reads:

"The UV absorption band in the presence of an aromatic nuclei (no other functional group specified) can be around 255 nm (Pretsch et al., 2000; Samburova et al., 2016), which is very close to our observed $\beta_{abs\_snow}$ values."

9) pg 12, line 19: The AAEs given in Doherty et al. (2010) are for 450-600nm. AAE over this wavelength band might be quite different from that at the band of 330-400nm given here.

The values for $AAE_{400}^{330}$ reported in our manuscript are from Zhang et al. (2019) who, in their Fig. 6, report $AAE_{400}^{330}$ and not $AAE_{600}^{450}$. We assume Zhang et al. (2019) have adapted the Doherty et al. (2010) values to the wavelength range of 330-400 properly.

Figure 1: Overall this is a useful figure. I was, however, a bit confused by the box on the left side that simply says "Sumlin et al. (2018)". What is this supposed to represent? If it's the source of the BrC size distribution, just put "Sumlin et al. (2018)" inside that box.

The Sumlin et al. (2018) study lends two items to our manuscript, the size distribution of BrC aerosol particles used in this study as well as values of derived BrC imaginary refractive index, which we use only to compare against values derived through our methods; hence, there is a "T" split in the figure. There are two dashed lines in shown in the legend as well, indicating that the dashed box emphasizing that "Sumlin et al. (2018)" is from an outside study. The figure caption has been updated to clarify the figure and the use of outside studies, and now reads as:

"Overview of methods used to derive BrC optical properties and compare measured and modeled albedo. The Sengupta et al. (2018) and Sumlin et al. (2018) studies both provide previously derived values of $\kappa_{BrC}$ to compare, and Sumlin et al. (2018) provide a size distribution of BrC aerosol under similar combustion conditions to those used in this study."

Figure 2: Can you add the radius/diameter of the deposition area to this figure, please?

Figure 2 has been updated to include the approximate radius of the deposition area and has been scaled a bit to better represent relative lengths, as seen here:

[Figure]

Figure 5 caption should state the R-squared at each wavelength (as shown in panel a for 275nm) is also given for each wavelength, and it should note at what wavelength resolution the mass-specific absorption is given.

Thank you for this suggestion; the figure caption has been updated and now reads:

"Panel (**a**): Absorption coefficient at 275 nm as function of total organic carbon (TOC) concentration. The fitted linear regression slope gives the mass-specific absorption $B_\lambda = 7.5 \pm 0.3$ m$^2$ g$^{-1}$ with a correlation coefficient of $R^2 = 0.98$. Panel (**b**): Mass-specific absorption, $B(\lambda)$, and $R^2$ for each value of $B(\lambda)$ across the wavelength range 215 – 815 nm for BrC deposited experimentally and found in natural snow at Tamarack Lake together at a 1-nm resolution. These panels include absorption coefficients of all meltwater samples characterized in Fig. 2."

Figure 7 caption should note that the real index of refraction is assumed to be a fixed value of 1.6 across all wavelengths. This is why both g and the mass extinction coefficient are so invariant (small gray area in panels b and c)!

Thank you for this suggestion; the figure caption has been updated to read:

"Single scattering albedo (SSA, Panel **a**), asymmetry parameter (g, Panel **b**), and mass extinction coefficient (MEC, m$^2$ kg$^{-1}$, Panel **c**) of BrC derived using Mie theory. BC optical properties used in the SNICAR model are provided for reference. The grey area represents the outputs of each parameter for the range of $\kappa_{BrC}$ retrievals used in the Mie code, where the largest relative discrepancy lies in the SSA. Note: the real refractive index for the calculations presented in this figure is assumed to be $n_\lambda = 1.6$ across all wavelengths."

Table 1: The air temp is given (footnote!) at a site 0.9km SE of the sampling site and at 2683m altitude. What is the altitude of the sampling site itself? Considerably different from 2683m? (Can you use the dry adiabatic lapse rate to estimate the sampling site temp if it wasn't measured, which would have been easy enough to do!)

The Tamarack Lake experiment site lies at an altitude of 2694m above sea level. The authors believe that the difference in altitude (~11 m) and linear distance (< 1 km) between the Mt. Rose SNOTEL site and the Tamarack Lake site does not represent a significant change in atmospheric or meteorological conditions; thus, the air temperatures provided are representative.

SUGGESTED SMALL EDITS:

Pg. 2, line 10: "cryosphere are a growing" –> "cryosphere is a growing"

Done; the statement now reads (Page 2, Lines 10-11):

> "The light-absorbing properties of BrC – an optically defined component of organic carbon (OC) aerosol – in the cryosphere is a growing topic of research."

Pg. 2, lines 20-22: Dang and Hegg (2014) should also be cited here.

Added. The statement now reads (Page 2, Lines 20-22):

> "Many studies have focused on the light absorption of the water-soluble fraction of OC, particularly that of humic-like substances (HULIS) (Dang and Hegg, 2014; Graber and Rudich, 2006; Samburova et al., 2005; Sun et al., 2007)."

Pg. 3, line 3: "toward shorter wavelength" –> "towards shorter wavelengths"

Done; the statement now reads (Page 3, Lines 2-4):

> "The imaginary part of the BrC refractive index (i.e., $\kappa\_\lambda$) increases greatly toward shorter wavelengths in the visible and near-UV spectrum, giving BrC its colored appearance and namesake (Andreae and Gelencsér, 2006).

Pg. 3, line 25: add a comma after "(Ervens et al., 2004)"

Done; the statement now reads (Page 3, Lines 24-25):

> "Chemical transformation of BrC during atmospheric transport can involve either fragmentation processes, producing lower molecular weight compounds (Ervens et al., 2004), or oligomerization (Carlton et al., 2007)."

Pg. 4, line 26: "presented in Fig. 1 to guide through the relationships" –> "presented in Fig. 1 as a guide to the relationships"

Done; the statement now reads (Page 4, Lines 26-27):

> "An overview of methods used for this work is presented in Fig. 1 as a guide to the relationships between measurements and modeling presented in the following sections."

Pg. 7, line 24: "This method assumes there is little contribution to scattering to the overall extinction of light along the path." These samples surely include insoluble particles, so there was some contribution from particulate scattering to the signal. Was the possible magnitude of this effect estimated? Would this constitute a positive bias in derived absorption?

The authors agree that the wording of this sentence may be a bit misleading in our intentions and assumptions made during our analyses. Indeed, the contribution to the scattered signal from insoluble particles is not quantified but is assumed to be low enough and present in both the snow albedo measurements after deposition and the absorbance measurements of the melted samples. Thus, any bias that is present in the measurements derived from scattered light is (assumed to be) accounted for. Additionally, visual inspection of the melted snow samples did not show any cloudiness, which would indicate scattering by a large fraction of insoluble particles. Instead, the uncertainty pertaining to insoluble particulates is present in the TOC analysis, where some insoluble OC may be unaccounted for (addressed in Section 4). We have updated the manuscript on Page 8, Line 1 to better state this assumption; it now reads:

> "This method assumes there is little contribution of scattering to the overall extinction of light along the path (Bosch et al., 2014). This is appropriate for a suspension or solution where extinction is

dominated by absorption caused by dissolved OC and/or by carbonaceous particles smaller than the wavelength of incident light, where extinction due to scattering can be neglected."

Pg. 9, line 25: ". . . in Fig. 3 alongside measured" –> ". . . in Fig. 3 alongside the measured"

Fixed. The updated sentence now reads (Page 10, Lines 7-9):

"This strong wavelength dependence was quantified by the measured spectral albedo for each deposition experiment, which is shown in Fig. 3 alongside the measured albedo of the natural snowpack upwind of the deposition."

Pg. 9, line 27: "In the UV, there was a stronger reduction than in the visible, up to ~0.14 and ~0-.21 at 350nm for Experiment 1 and 2, respectively (Fig. 3)" –> "In the UV, there was a stronger reduction in albedo than in the visible, of ~0.14 and ~0-.21 at 350nm for Experiment 1 and 2, respectively, relative to the natural snowpack albedo (Fig. 3)"

Fixed. The updated sentence now reads (Page 10, Lines 10-12):

"In the UV, there was a stronger reduction in albedo than in the visible, of ~0.14 and ~0.21 at 350 nm for Experiment 1 and 2, respectively, relative to the natural snowpack albedo (Fig. 3)."

Pg. 10, line 23: "snow sampling sites, L1 TOC" –> "snow sampling sites, the L1 TOC"

Added "the". The new sentence now reads (Page 11, Lines 2-4):

"Averaging samples collected at both natural snow sampling sites, the L1 TOC concentration was $0.579 \pm 0.014$ g m$^{-3}$, for L2 it was $0.436 \pm 0.048$ g m$^{-3}$, and for L3 it was $0.425 \pm 0.008$ g m$^{-3}$."

Pg. 10, line 26: "air pollution from mining. . ." –> "air pollution from the mining. . ."

Done. The sentence now reads (Page 11, Lines 5-7):

"Meinander et al. (2013) found a factor of six larger, where concentrations are heavily influenced through deposition of air pollution from the mining and refining industry."

Pg. 10, line 30: "concentration than Experiment 1" –> "concentration as Experiment 1"

Fixed. The sentence now reads (Page 11, Lines 9-11):

"Experiment 2 deposited approximately twice the TOC mass concentration as Experiment 1, which is evidenced not only in the measured TOC concentration for each layer, but also in a stronger decrease of measured UV albedo (Fig. 3)."

Pg. 10, lines 31-32: delete ". . .for Experiments 2 than for Experiment 1 relative to the natural snow" (it's not needed).

Done. The new sentence now reads (Page 11, Lines 9-11):

"Experiment 2 deposited approximately twice the TOC mass concentration as Experiment 1, which is evidenced not only in the measured TOC concentration for each layer, but also in a stronger decrease of measured UV albedo (Fig. 3)."

Pg. 11: "UV absorption band in presence of an aromatic.." –> "The UV absorption band in The presence of an aromatic.."

Added "the", and the updated manuscript now reads (Page 11, Line 24):

"The UV absorption band in the presence of an aromatic nuclei (no other functional group specified) can be around 255 nm (Pretsch et al., 2000; Samburova et al., 2016), which is very close to our observed $\beta_{abs\_snow}$ values."

pg. 11, lines 29-30: I'm not sure what you're trying to say here with ". . .together with those of the nature snowpack, TOCsnow, as one dataset." Figure 5 only shows one dataset. ?

Reference to "one dataset" refers to the fact that the TOC concentrations from the natural snow and TOC concentrations from the deposition area snow are used together in the linear regression to derive the value of the specific absorbance, $B_\lambda$, instead of deriving a value of $B_\lambda$ for each individual set of TOC concentrations: one for $TOC_{snow}$ and one for $TOC_{BrC}$. This strengthens the derivation of $B_\lambda$ over the wavelength range to include more TOC/absorbance pairs for higher concentrations.

pg. 13, line 24: Figure 7 should be referenced at the end of the sentence at the beginning of this line.

Added. The new sentence now reads (Page 14, Lines 2-4):

[revised manuscript text omitted]

**Reply to the review by handling editor Dr. Timothy Garrett and overview of other minor changes for the manuscript, "Deposition of Brown Carbon onto Snow: changes of snow optical and radiative properties" by N. D. Beres et al.**

We thank Dr. Garrett for his suggested adjustment to the manuscript. Dr. Garrett's comment is in blue and the responses by the manuscript authors are in black.

One adjustment I request is that a link is provided to the datasets placed a formal data repository, replacing the statement "Data will be made available upon request by the corresponding author".

The data presented throughout the manuscript has been uploaded to the following Zenodo respository:

https://doi.org/10.5281/zenodo.3736325

Data was uploaded in the form of an Excel spreadsheet, where each tab of the spreadsheet corresponds to the data presented in the figures and tables throughout the manuscript. Additionally, the "Data Availability" statement at the end of the manuscript was replaced to read (Page 18, Lines 25-26):

> "*Data Availability:* Data presented and used throughout this study can be accessed through the following data repository: https://doi.org/10.5281/zenodo.3736325."

Additionally, one update to a figure was implemented. The panels (**a**) and (**b**) of Figure 3 were arranged in the wrong order. Before, the figure was shown as:

[Figure]

They have been updated to reflect the correct order of the experiment and to match the caption, and are now shown as:

[revised manuscript text omitted]